# Region-specific regulation of stem cell-driven regeneration in tapeworms

**Tania Rozario[1]\*, Edward B Quinn[1], Jianbin Wang[2], Richard E Davis[2], Phillip A Newmark[1,3,4]\***

[1]Morgridge Institute for Research, Madison, United States; [2]RNA Bioscience Initiative, Department of Biochemistry and Molecular Genetics, University of Colorado School of Medicine, Aurora, United States; [3]Howard Hughes Medical Institute, Chevy Chase, United States; [4]Department of Integrative Biology, University of Wisconsin–Madison, Madison, United States

**Abstract** Tapeworms grow at rates rivaling the fastest-growing metazoan tissues. To propagate they shed large parts of their body; to replace these lost tissues they regenerate proglottids (segments) as part of normal homeostasis. Their remarkable growth and regeneration are fueled by adult somatic stem cells that have yet to be characterized molecularly. Using the rat intestinal tapeworm, *Hymenolepis diminuta*, we find that regenerative potential is regionally limited to the neck, where head-dependent extrinsic signals create a permissive microenvironment for stem cell-driven regeneration. Using transcriptomic analyses and RNA interference, we characterize and functionally validate regulators of tapeworm growth and regeneration. We find no evidence that stem cells are restricted to the regeneration-competent neck. Instead, lethally irradiated tapeworms can be rescued when cells from either regeneration-competent or regeneration-incompetent regions are transplanted into the neck. Together, the head and neck tissues provide extrinsic cues that regulate stem cells, enabling region-specific regeneration in this parasite.
DOI: https://doi.org/10.7554/eLife.48958.001

\*For correspondence:
trozario@morgridge.org (TR);
pnewmark@morgridge.org (PAN)

## Introduction

Tapeworms are parasitic flatworms that infect humans and livestock, causing lost economic output, disease, and in rare cases, death (*Del Brutto, 2013*). These parasites are well known for their ability to reach enormous lengths. For example, humans infected with the broad or fish tapeworm, *Diphyllobothrium latum*, harbor parasites that average 6 m in length (*Craig and Ito, 2007*). It is less commonly appreciated that tapeworms can regenerate to accommodate their multi-host life cycle. Adult tapeworms in their host intestines develop proglottids (segments) that are gravid with embryos. Tapeworms pinch off the posterior and gravid sections of their body, which exit with the host excrement, to be eaten by a suitable intermediate host that supports larval tapeworm development. Despite losing large body sections, the tapeworm does not progressively shorten; instead, it regenerates proglottids, allowing the worms to maintain an equilibrium length. Despite this remarkable biology, tapeworms are an unexplored animal model in the study of regenerative behaviors.

Up to the 1980s the rat intestinal tapeworm, *Hymenolepis diminuta*, had been a favorite model organism among parasitologists. *H. diminuta* grows rapidly–within the first 15 days of infection, it produces up to 2200 proglottids, increases in length by up to 3400 times, and weight by up to 1.8 million times (*Roberts, 1980*)–and is easily propagated in the laboratory. Foundational work on their biochemistry, ultrastructure, and developmental biology enriched our understanding of these tapeworms (*Arai, 1980*). However, with the dawn of the molecular age and the rise of genetic model organisms, *H. diminuta* was essentially left behind. Here, we show that *H. diminuta* is an excellent,

**eLife digest** Many worms have remarkable abilities to regrow and repair their bodies. The parasitic tapeworms, for example, can reach lengths of several meters and grow much more quickly than tissues in humans and other complex animals. This growth allows tapeworms to counteract the continual loss of the segments that make up their bodies, known as proglottids – a process that happens throughout their lives.

The capacity to regenerate thousands of lost body segments and maintain an overall body length suggests that tapeworms have groups of stem cells in their body which can grow and divide to produce the new body parts. Yet, regeneration in tapeworms has not been closely studied.

Rozario et al. have now examined *Hymenolepsis diminuta*, the rat tapeworm, and identified the neck of the tapeworm as crucial for its ability to regrow lost body segments. Further analysis identified two genes, *zmym3* and *pogzl*, that are essential for cell division during tapeworm growth. However, Rozario et al. showed that these genes are active elsewhere in the worm's body and that it is the conditions found specifically in the tapeworm's neck that create the right environment for stem cells to enable regeneration of new segments.

Tapeworms provide a valuable example for studying the growth of stem cells and these findings highlight the important role that the cells' surroundings play in driving stem cell activity. These findings could also lead to new insights into how stem cells behave in other animals and could potentially lead to new approaches to prevent or treat tapeworm infections.

DOI: https://doi.org/10.7554/eLife.48958.002

tractable model for the study of stem cells and regeneration, with the power to inform us about parasite physiology.

As an obligate endoparasite, adult *H. diminuta* will expire once its host rat dies. However, the lifespan of *H. diminuta* can be greatly increased via regeneration. A single adult tapeworm can be serially amputated and transplanted into a new host intestine, where the fragment can regenerate into a mature tapeworm even after 13 rounds of amputation over 14 years (*Read, 1967*). These observations have led to speculation that *H. diminuta* may be inherently immortal. This situation is reminiscent of the free-living cousins of tapeworms: freshwater planarians like *Schmidtea mediterranea*, which reproduce indefinitely by fission, and can regenerate their whole body from tiny fragments (*Newmark and Sánchez Alvarado, 2002*).

Planarian immortality and regeneration are enabled by adult somatic stem cells called neoblasts (*Newmark and Sánchez Alvarado, 2002*; *Reddien, 2018*; *Baguñà, 2012*). These stem cells are the only dividing undifferentiated cells within the soma. Like planarians, *H. diminuta* maintains a population of neoblast-like adult somatic stem cells (*Roberts, 1980*) that are likely responsible for their growth and regenerative ability. Recently, stem cells of multiple species of parasitic flatworms have been described (*Collins et al., 2013*; *Koziol et al., 2014*; *Koziol et al., 2015*; *Wang et al., 2013*; *Koziol et al., 2010*). Stem cells play crucial roles in parasite development, transmission, homeostasis, and even disease. For example, stem cells enable prolific reproduction and longevity (*Collins, 2017*), mediate host-parasite interactions (*Collins et al., 2016*), and allow metastatic parasite transmission in host tissues (*Brehm and Koziol, 2014*). How stem cells may regulate regeneration in parasites such as tapeworms is largely unexplored and the subject of this study.

We use *H. diminuta,* to investigate the molecular basis of tapeworm regeneration. We have established and refined experimental tools such as transcriptomics, in vitro parasite culture, whole-mount and fluorescent RNA in situ hybridization (WISH and FISH), cycling-cell tracing with thymidine analogs, RNA interference (RNAi), and cell transplantation, all described in this work. We determine that the ability to regenerate is regionally limited to the neck of adult *H. diminuta*. However, regeneration from the neck is finite without signals from the tapeworm head. Using RNA sequencing (RNA-seq), we identify and characterize various markers of the somatic cycling-cell population, which includes tapeworm stem cells. Using RNAi, we functionally validate molecular regulators of growth and regeneration. However, our analyses failed to uncover a neck-specific stem cell population that explains the regional regenerative ability displayed by *H. diminuta*. Instead, we show that cells from both regeneration-competent and regeneration-incompetent regions of *H. diminuta* have stem cell

ability and can restore viability to lethally irradiated tapeworms. Our results show that extrinsic signals present in the tapeworm neck, rather than specialized stem cells, confer region-specific regenerative ability in this tapeworm.

## Results

The anatomy of adult *H. diminuta* consists of a head with four suckers, an unsegmented neck, and a body with thousands of proglottids/segments that grow and mature in an anterior-to-posterior direction (*Roberts, 1980*; *Rozario and Newmark, 2015*) (*Figure 1a*). What regions of the tapeworm body are competent to regenerate? In order to test regeneration competency, it is necessary to grow tapeworms in vitro instead of in the intestine, where the suckers are required to maintain parasites in vivo. We established *H. diminuta* in vitro culture conditions modified from Schiller's method (*Schiller, 1965*) and tested the regeneration competence of 1 cm amputated fragments (*Figure 1b–c*). The anterior-most fragments (head+neck+body) were competent to regenerate, confirming in vivo observations using amputation and transplantation (*Read, 1967*; *Goodchild, 1958*). Anterior fragments that were first decapitated (neck+body) were also competent to regenerate. In contrast, 'body only' fragments failed to regenerate proglottids. All amputated fragments could grow in length (*Figure 1d*), differentiate mature reproductive structures, and mate. Despite the failure to regenerate, 'body only' fragments could grow because each existing proglottid increased in length as it progressively matured (*Figure 1—figure supplement 1a–b*). However, only fragments that retained the neck were able to regenerate new proglottids over time. The neck of 6-day-old tapeworms used in this study is typically 2–3 mm long when observed after DAPI staining and widefield fluorescent microscopy. By amputating 2 mm 'neck only' fragments, we find that the neck is sufficient to regenerate an average of 383 proglottids (SD = 138, N = 4, n = 20) after 12 days in vitro (*Figure 1e*). In no case did we observe head regeneration. Furthermore, amputated heads alone could not regenerate in vitro (*Figure 1—figure supplement 1c*) nor in vivo (*Read, 1967*). Thus, neither the head nor body can regenerate proglottids, but the neck is both necessary and sufficient for proglottid-specific regeneration in *H. diminuta*.

Previous in vivo studies have shown that *H. diminuta* can regenerate after serial rounds of amputation and transplantation for over a decade (*Read, 1967*) and perhaps indefinitely. Using in vitro culture, we confirmed that anterior fragments of *H. diminuta* can regenerate after at least four rounds of serial amputation (*Figure 1f–g*). Decapitated (-head) fragments regenerated proglottids after the first amputation; however, re-amputation abrogated regeneration (*Figure 1f–g*). After decapitation, a definitive neck could not be maintained and eventually, the whole tissue was comprised of proglottids (*Figure 1—figure supplement 2*). Without the head, proglottid regeneration from the neck is finite. Thus, while the neck is necessary and sufficient for proglottid regeneration, the head is required to maintain an unsegmented neck and for persistent regeneration.

If signals from the head regulate regeneration, is regenerative potential asymmetric across the anterior-posterior (A-P) axis of the neck? We subdivided the neck into three 1 mm fragments and found that the most-anterior neck fragments regenerated more proglottids than the middle or posterior neck fragments (*Figure 1h–i*). Thus, regeneration potential is asymmetric across the neck A-P axis with a strong anterior bias.

Since the neck is the only region competent to regenerate, are stem cells preferentially confined to the neck? In lieu of specific molecular markers for stem cells, we examined the distribution of all cycling cells in adult tapeworms. In flatworms, it has been repeatedly shown that the only proliferative somatic cells are undifferentiated cells with stem cell morphology and/or function; these cells have been termed neoblasts, adult somatic stem cells, or germinative cells, depending on the organism (*Collins et al., 2013*; *Koziol et al., 2014*; *Baguñà et al., 1989*; *Newmark and Sánchez Alvarado, 2000*; *Ladurner et al., 2000*). In *H. diminuta*, proliferation does not occur in regions comprised solely of differentiated cells (muscle and tegument/parasite skin at the animal edge) (*Bolla and Roberts, 1971*). Instead, proliferation is only detected in regions where undifferentiated cells with the typical morphology of stem cells can be distinguished (*Bolla and Roberts, 1971*; *Sulgostowska, 1972*). Thus, cycling somatic cells in *H. diminuta* would not include differentiated cells, but would include stem cells and any dividing progeny. To label cycling cells, we used two methods: (i) uptake of the thymidine analog F-*ara*-EdU (*Neef and Luedtke, 2011*) to mark cells in S-phase and (ii) FISH to detect cell cycle-regulated transcripts, such as the replication licensing factor

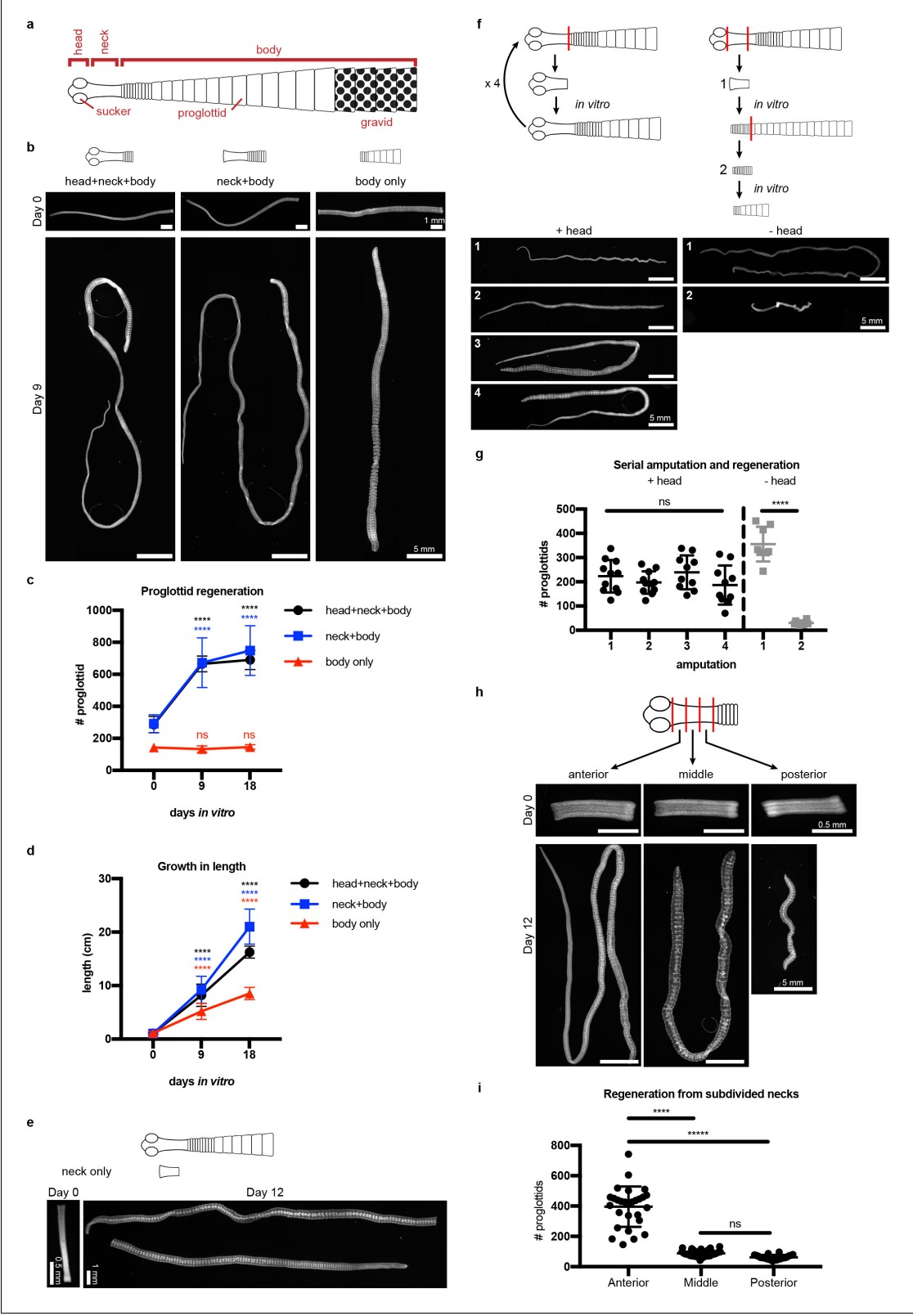

**Figure 1.** Regeneration competence of *H. diminuta*. (**a**) Schematic of *H. diminuta* adults. (**b**) DAPI-stained 1 cm fragments grown in vitro. (**c–d**) Quantification of proglottid number and growth in length from (**b**). Error bars = SD, N = 2–5, n = 7–21; one-way ANOVA with Dunnett's multiple comparison test, compared to day 0. (**e**) Representative DAPI-stained 'neck only' fragment regeneration. (**f–g**) 2 mm anterior fragments, with or without the head, grown in vitro for 12–15 days and then re-amputated serially. Error bars = SD, +head: one-way ANOVA with Tukey's multiple comparison

*Figure 1 continued on next page*

*Figure 1 continued*

test, -head: Student's t-test. (**h–i**) DAPI-stained 1 mm fragments from the anterior, middle, and posterior of the neck grown in vitro. Error bars = SD, N = 3, n = 22–29, one-way ANOVA with Tukey's multiple comparison test.

DOI: https://doi.org/10.7554/eLife.48958.003

The following figure supplements are available for figure 1:

**Figure supplement 1.** Phenotypic description of regeneration-incompetent tissues of *H. diminuta*.
DOI: https://doi.org/10.7554/eLife.48958.004

**Figure supplement 2.** Unsegmented neck is depleted after decapitation.
DOI: https://doi.org/10.7554/eLife.48958.005

*minichromosome maintenance complex component 2* (*mcm2*) and *histone h2b* (*h2b*), which are conserved cycling-cell markers in free-living and parasitic flatworms (*Collins et al., 2013*; *Solana et al., 2012*). We detected cycling somatic cells throughout the tapeworm body (*Figure 2a–b*). Contrary to previous results (*Bolla and Roberts, 1971*), we also detected cycling cells in the head, though in

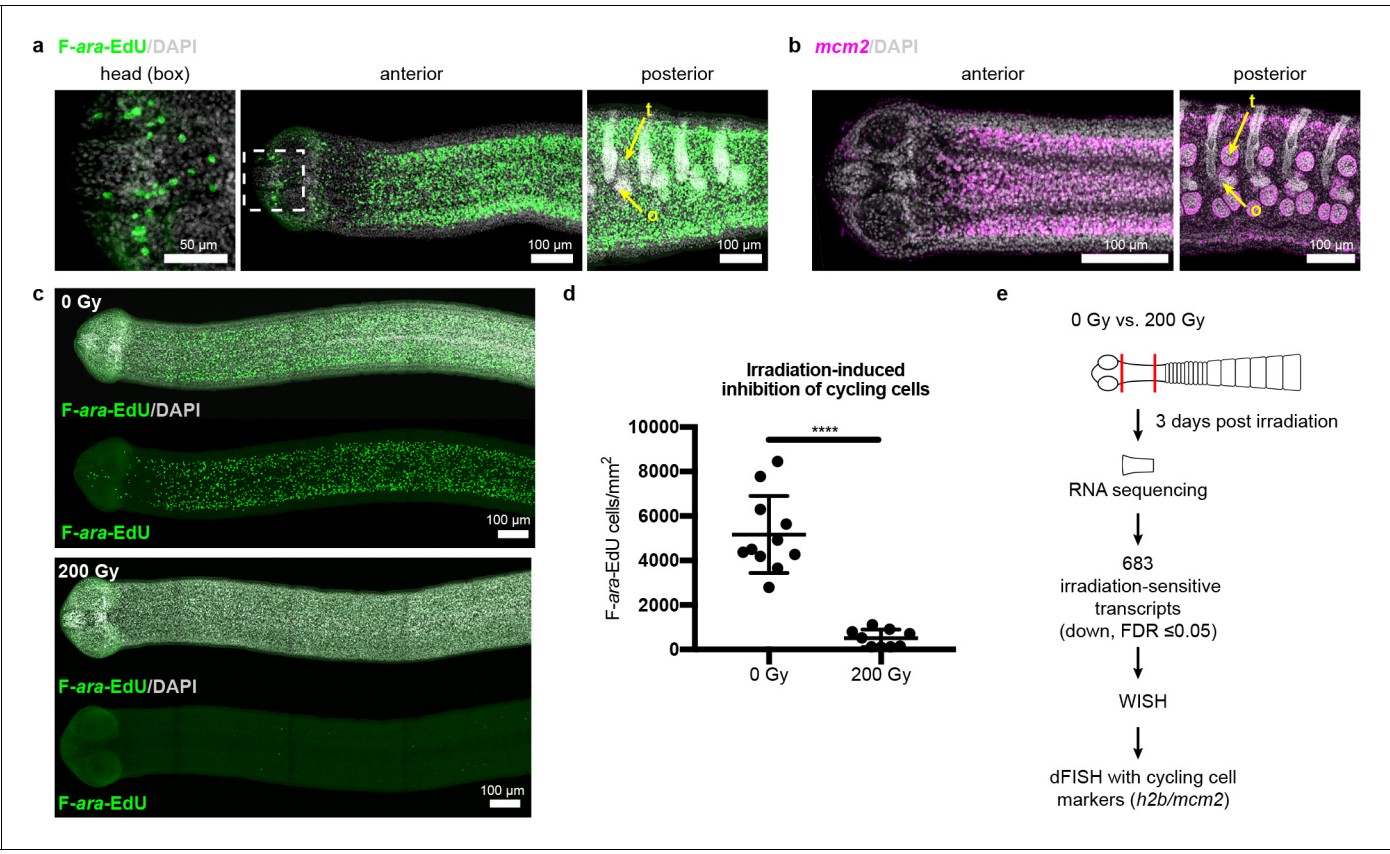

**Figure 2.** Cycling somatic cells are distributed throughout the tapeworm body and are irradiation sensitive. (**a-b**) Maximum-intensity projections of confocal sections showing distribution of cycling cells by 2 hr uptake of F-*ara*-EdU (**a**) or FISH for *mcm2* (**b**). Fewer cycling cells were found in the head (box), while abundant cycling cells were observed in both somatic and gonadal tissues throughout the body. t = testis, o = ovary. (**c**) Maximum-intensity projections of tile-stitched confocal sections after 1 hr uptake of F-*ara*-EdU (green) 3 days post-irradiation. (**d**) Quantification of F-*ara*-EdU⁺ cell inhibition from (**c**). Error bars = SD, N = 2, n = 11 and 9, Student's t-test. (**e**) RNA-seq strategy to identify genes expressed in cycling cells. (Nuclei are counterstained with DAPI (gray) in this and all subsequent figures.).

DOI: https://doi.org/10.7554/eLife.48958.006

The following figure supplements are available for figure 2:

**Figure supplement 1.** Irradiation inhibits tapeworm regeneration.
DOI: https://doi.org/10.7554/eLife.48958.007

**Figure supplement 2.** Validation of RNA-seq by WISH after irradiation.
DOI: https://doi.org/10.7554/eLife.48958.008

small numbers (*Figure 2a*). The scarcity of these cells may be the reason they were originally missed. Taken together, cycling cells are present in all regions, regardless of regeneration competence.

To further our understanding of how tapeworm stem cells are distributed and regulated, we sought to identify stem cell markers. Stem cell genes have been discovered previously in flatworms by identifying transcripts downregulated after exposure to irradiation, which depletes cycling cells (*Collins et al., 2013*; *Solana et al., 2012*; *Eisenhoffer et al., 2008*). Exposing *H. diminuta* to 200 Gy x-irradiation reduced the number of cycling cells by 91 ± 6% after 3 days (*Figure 2c–d*) and abrogated both growth and regeneration (*Figure 2—figure supplement 1a–b*). This dosage is lethal; all fragments from worms exposed to 200 Gy x-irradiation degenerated after 1 month (*Figure 2—figure supplement 1c–d*). We leveraged the sensitivity of *H. diminuta* to lethal irradiation in order to identify new molecular markers of cycling somatic cells by RNA-seq (*Figure 2e*). A de novo transcriptome of 14,346 transcripts was assembled (see Materials and methods) to which sequencing reads were mapped. We identified 683 transcripts that were irradiation sensitive (downregulated; FDR $\leq$ 0.05) (*Supplementary file 1a*). Expression of irradiation-sensitive transcripts by WISH was indeed reduced after exposure to irradiation, validating our RNA-seq approach (*Figure 2—figure supplement 2*).

Two rounds of expression screening were then applied to hone in on cycling-cell transcripts from our irradiation-sensitive dataset (*Figure 2e*). The position of cycling cells in the neck is spatially restricted in a conserved pattern (*Koziol and Castillo, 2011*) (*Figure 3a*): cycling cells reside in the neck parenchyma bounded by the nerve cords and are absent from the animal edge where muscle and tegument are located (*Bolla and Roberts, 1971*). Among 194 irradiation-sensitive transcripts that displayed clear WISH patterns, 63% were expressed in the neck parenchyma, though in a variety of patterns (*Figure 3—figure supplement 1*). 13% showed similar patterns to *h2b* and *mcm2* (*Figure 3b–c*, *Figure 3—figure supplement 1b*). These include the predicted nucleic acid binding factors *Zn finger MYM type 3* (*zmym3*) and *pogo transposable element with ZN finger domain-like* (*pogzl*), as well as *NAB co-repressor domain two superfamily member* (*nab2*) and nuclear lamina component *laminB receptor* (*lbr*). 25% of irradiation-sensitive transcripts, were expressed in a minority of cells in the neck parenchyma (*Figure 3—figure supplement 1c*). 24% were expressed within the parenchyma and more broadly toward the animal edge (*Figure 3—figure supplement 1d*). The remainder represented transcripts expressed at segment boundaries or in differentiated tissues (*Figure 3—figure supplement 1e–f*). All transcripts that were expressed in the neck parenchyma were also found throughout the worm body, even in the most posterior proglottids (*Figure 3—figure supplement 1b–c*). In conclusion, irradiation-sensitive transcripts identified by RNA-seq likely represent markers for stem cells, progenitors, and even differentiated cells that were lost or compromised following irradiation.

To focus on transcripts with enriched expression in cycling cells, we performed double FISH (dFISH) with irradiation-sensitive candidates and either *h2b* or *mcm2,* which we used interchangeably as they are co-expressed in the neck parenchyma (*Figure 3—figure supplement 2*). After dFISH for 53 candidates, 72% of transcripts tested were co-expressed in cycling cells (*Figure 3—figure supplement 3a*, *Supplementary file 1b*). The irradiation-sensitive transcripts from *Figure 3c* were indeed colocalized in cycling somatic cells (*Figure 3d*). One transcript, the homeobox factor *prospero* (*prox1*), was expressed exclusively in a subset of cycling cells (*Figure 3—figure supplement 3b*). We confirmed that genes with expression that only partially overlapped in the neck parenchyma, such as the Zn finger-containing gene *HDt_10981* and *palmitoyl-protein thioesterase 1* (*ppt1*), were expressed in both cycling cells and non-cycling cells (*Figure 3—figure supplement 3c*). We propose that these genes likely represent lineage-committed stem cells or progenitors for tissues such as muscle, neurons, tegument, or protonephridia. 28% of irradiation-sensitive transcripts were predominantly expressed in non-cycling cells that were juxtaposed to cycling cells (*Figure 3—figure supplement 3d*). The transcriptional heterogeneity detected in the cycling-cell compartment is reminiscent of similar observations made in the regenerative planarian *S. mediterranea* (*Reddien, 2018*). A comparative analysis between verified tapeworm cycling-cell transcripts and their putative planarian homologs revealed a number of transcripts with conserved expression in cycling-cell populations from these distantly related flatworms (*Supplementary file 1c*) (see Discussion). In summary, our analysis revealed a heterogeneous and complex mixture of cell types or states in the neck parenchyma as well as within the cycling-cell population.

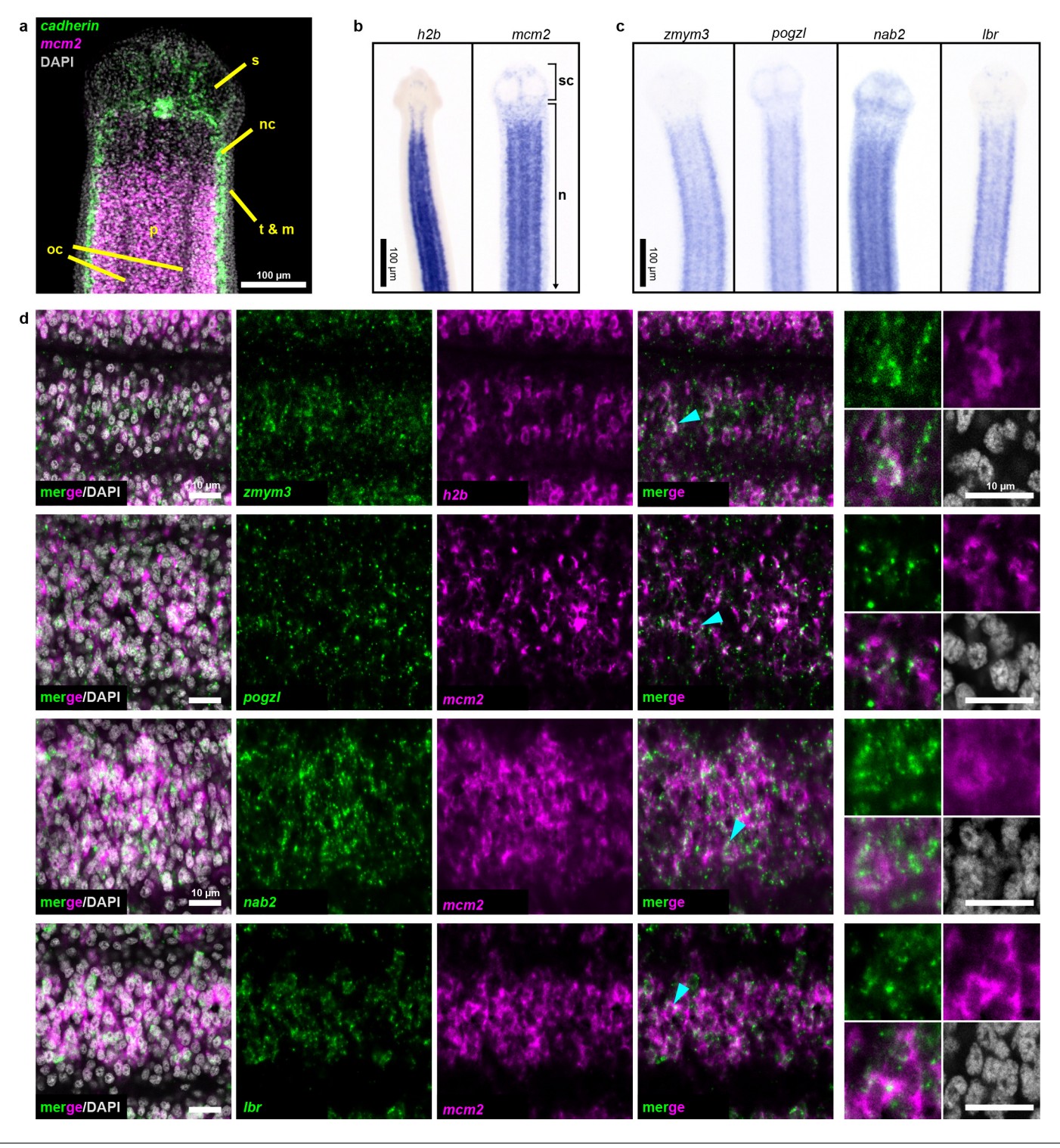

**Figure 3.** Expression screening for cycling cell markers. (a) Confocal section of a tapeworm anterior. Cycling cells (*mcm2*: magenta) in the neck parenchyma are between the nerve cords (*cadherin*: green). s: sucker, nc: nerve cord, oc: osmoregulatory canal, t: tegument, m: muscle, and p: parenchyma. (b) WISH of known cycling-cell markers *h2b* and *mcm2*. sc: scolex (head) and n: neck. (c) WISH for irradiation-sensitive transcripts expressed in the neck parenchyma. (d) Confocal sections of dFISH for irradiation-sensitive transcripts (green) with *h2b* or *mcm2* (magenta) from neck parenchyma. Cyan arrowheads indicate cells magnified at the far right.

DOI: https://doi.org/10.7554/eLife.48958.009

*Figure 3 continued on next page*

*Figure 3 continued*

The following figure supplements are available for figure 3:

**Figure supplement 1.** WISH patterns of irradiation-sensitive transcripts identified using RNA-seq.
DOI: https://doi.org/10.7554/eLife.48958.010
**Figure supplement 2.** Coexpression of *mcm2* and *h2b.*
DOI: https://doi.org/10.7554/eLife.48958.011
**Figure supplement 3.** The cycling somatic cell population is heterogeneous.
DOI: https://doi.org/10.7554/eLife.48958.012

What role(s) do the newly identified cycling-cell genes play during regeneration? We performed RNAi of target genes, confirmed knockdown by quantitative PCR (*Figure 4—figure supplement 1*), and assayed for defects in growth and regeneration (*Figure 4a*). As a proof of principle, we knocked down *h2b,* which should compromise growth due to cycling cell loss, as observed in other flatworms (*Collins et al., 2016*; *Solana et al., 2012*). Knockdown of *h2b, zmym3,* and *pogzl* each resulted in diminished growth and regeneration (*Figure 4b–c*). The number of proglottids regenerated was also reduced, but could not be quantified as many RNAi worms were so thin and frail (*Figure 4b*) that proglottid definition was lost.

Are these RNAi-induced failures in growth and regeneration due to defects in the cycling-cell population? RNAi knockdown of *h2b, zmym3,* and *pogzl* severely reduced the number of proliferative cells in the neck that could incorporate F-*ara*-EdU (*Figure 4d–e*). We also observed fewer *mcm2*$^+$ cells after RNAi (*Figure 4f*). Taken together, these results indicate that the cycling-cell population is either lost or dysregulated. Therefore, *h2b, zmym3,* and *pogzl* are necessary for the maintenance and/or proper function of cycling cells, likely including stem cells, in *H. diminuta*.

Although we have identified heterogeneity within the cycling-cell population of the neck parenchyma and uncovered genes that are required for growth and regeneration, it remains unclear why regeneration competence is restricted to the neck. By WISH and FISH, all cycling-cell transcripts including *zmym3* and *pogzl* were detected throughout the whole tapeworm body (*Figure 4g*, *Figure 3—figure supplement 1b–c*). In the tapeworm posterior, *zmym3* and *pogzl* were expressed in gonadal tissues (which contain mitotic germ cells) but also in somatic cells within the parenchyma (*Figure 4—figure supplement 2*). If *zmym3* and *pogzl* mark stem cells, this suggests that stem cells reside even in posterior tissues that are not competent to regenerate. Since *zmym3* and *pogzl* label all cycling cells, it is possible that stem cells of limited potential exist in the posterior, but an elusive subpopulation of pluripotent stem cells is confined to the neck.

Since we observed an anterior bias in regenerative ability (*Figure 1h–i*), we hypothesized that RNA-seq may reveal an anteriorly biased stem cell distribution that may point us to a pluripotent stem cell subpopulation. Thus, we performed RNA-seq of 1 mm anterior, middle, and posterior neck fragments (*Figure 1h*), and identified 461 anterior-enriched and 241 anterior-depleted transcripts (*Supplementary file 1d*). By WISH, anterior-enriched and anterior-depleted transcripts were often detected in corresponding gradients (*Figure 5a*), but in patterns that were excluded from the neck parenchyma. When we overlaid the anterior-enriched and -depleted datasets with our irradiation-sensitive dataset, the majority of anterior-enriched transcripts (88%) were not irradiation sensitive (*Figure 5b*). Our results suggest that the A-P polarized signals across the neck region arise predominantly within the non-cycling compartments.

Since our RNA-seq analysis identified 57 transcripts that were anterior enriched and irradiation sensitive, we examined expression patterns within this category. We found 15 transcripts expressed in a subset of cells within the neck parenchyma (*Figure 5c*) and initially hypothesized that these transcripts may represent subsets of stem cells. We were able to successfully test eight candidates by dFISH with cycling-cell markers and found that the majority (7/8) were not expressed in cycling cells (*Figure 5d*, *Supplementary file 1b*). Only *prox1* was co-expressed in cycling cells (*Figure 3—figure supplement 3b*). At present, the identity and function of *prox1*$^+$ cells is unknown. Furthermore, *prox1* is expressed throughout the tapeworm body (*Figure 3—figure supplement 1*). Thus, our analyses have not revealed an anteriorly biased subpopulation of stem cells that confer regenerative ability.

With no evidence for a unique neck-specific subpopulation of stem cells, we hypothesized that stem cells may be distributed throughout the tapeworm, but that extrinsic signals functioning in the

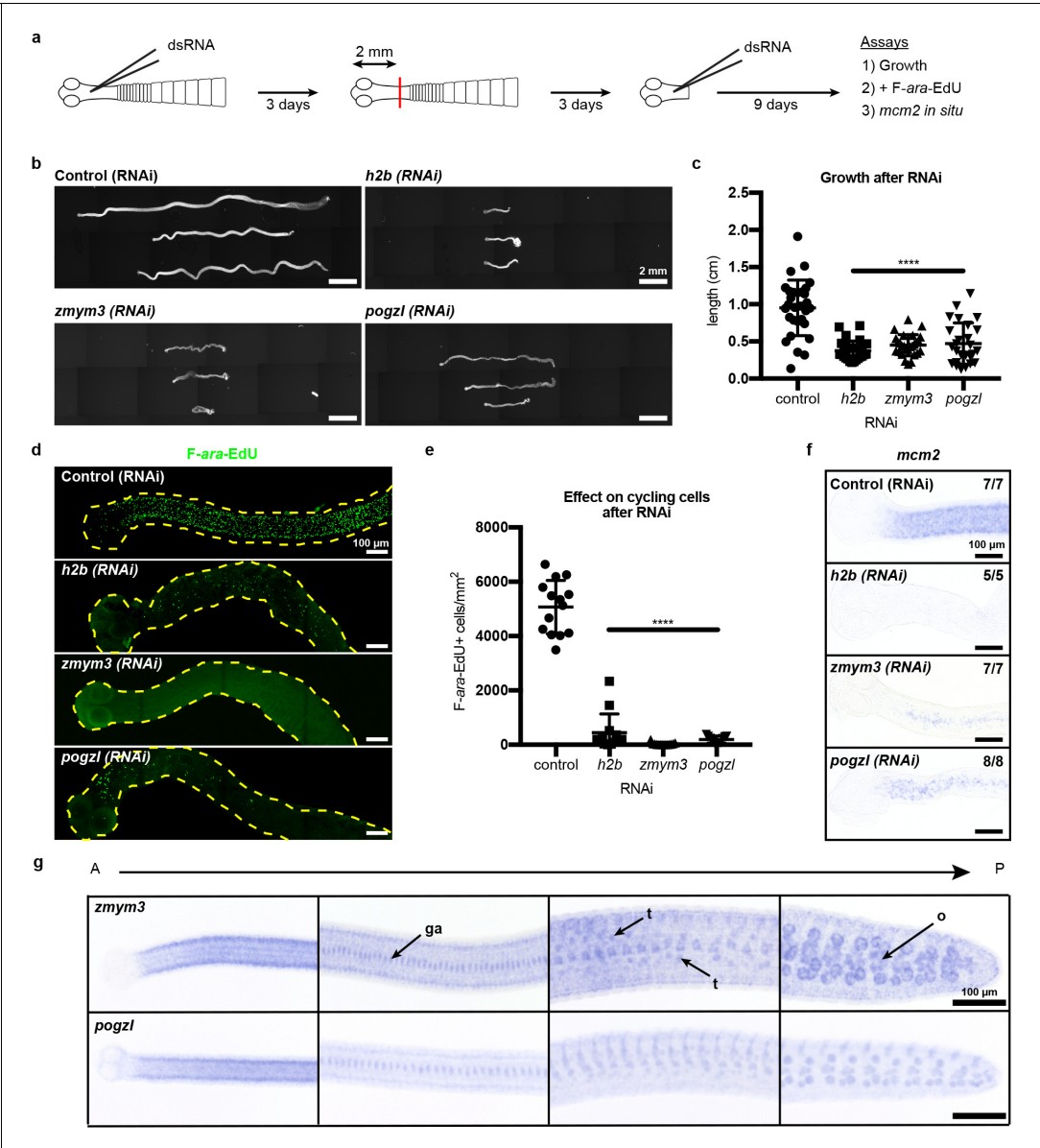

**Figure 4.** RNAi to identify genes required for growth and regeneration in *H. diminuta*. (**a**) Schematic of RNAi paradigm. (**b**) DAPI-stained worms after RNAi knockdown of *h2b*, *zmym3*, and *pogzl*. (**c**) Quantification of worm lengths after RNAi. Error bars = SD, N = 3–4, n = 26–37, one-way ANOVA with Dunnett's multiple comparison test compared to control. (**d-e**) Maximum-intensity projections (**d**) and quantification (**e**) of cycling-cell inhibition after 1 hr F-*ara*-EdU uptake following RNAi. Worms with degenerated necks were excluded from analysis. Error bars = SD, N = 3, n = 11–14, one-way ANOVA with Dunnett's multiple comparison test compared to control. (**f**) *mcm2* WISH on worm anteriors after RNAi. (**g**) WISH of *zmym3* and *pogzl* sampled from anterior to posterior of adult 6-day-old worms. ga: genital anlagen; t: testis; o: ovary.

DOI: https://doi.org/10.7554/eLife.48958.013

The following figure supplements are available for figure 4:

**Figure supplement 1.** Validation of target gene knockdown by quantitative PCR.
DOI: https://doi.org/10.7554/eLife.48958.014
**Figure supplement 2.** Expression of *zmym3* and *pogzl* in posterior proglottids.
DOI: https://doi.org/10.7554/eLife.48958.015

neck are necessary to instruct stem cell behavior and/or proglottid regeneration. We designed a functional assay to test populations of cells for the ability to rescue regeneration, modelled after similar experiments performed on planarians (*Baguñà, 2012*). We exposed tapeworms to a lethal dose of x-irradiation (200 Gy), injected cells from wild-type donors into the neck region, amputated 5 mm

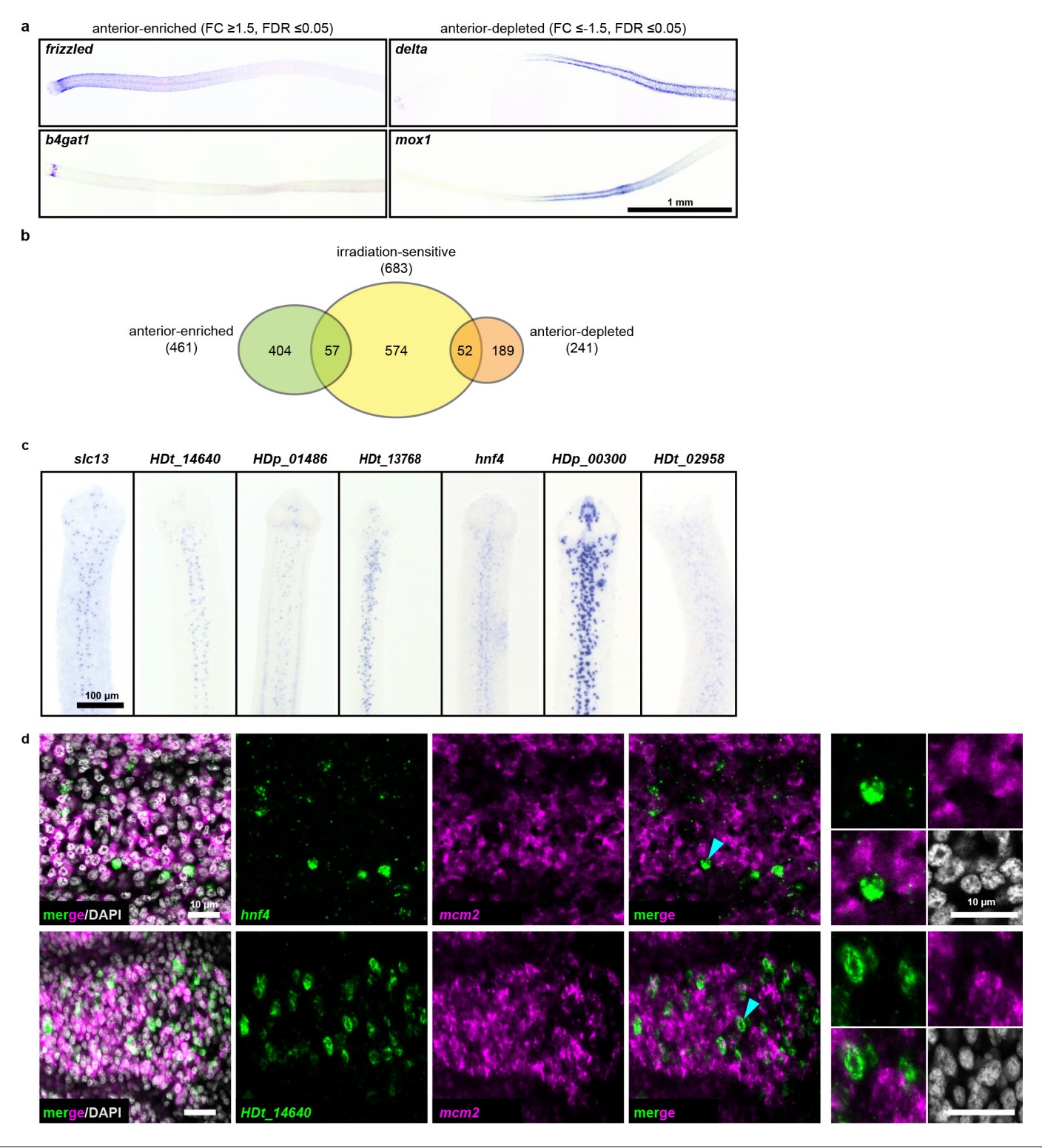

**Figure 5.** RNA-seq identifies anterior-enriched transcripts that are expressed predominantly in non-cycling cells. (a) WISH of tapeworm anteriors for transcripts that were anterior-enriched (FC ≥1.5, FDR ≤ 0.05) or -depleted (FC ≤−1.5, FDR ≤ 0.05) by RNA-seq. Panels oriented anterior facing left. (b) Differential gene expression analyses of 1 mm anterior, middle, and posterior neck fragments overlaid with irradiation-sensitive transcripts. (c) WISH of transcripts that were anterior-enriched and irradiation-sensitive by RNA-seq that showed expression in a subset of cells in the neck parenchyma. (d) Confocal sections from dFISH of anterior-enriched transcripts (green) and *mcm2* (magenta). Cyan arrowheads indicate cells that are magnified at the far right.

DOI: https://doi.org/10.7554/eLife.48958.016

anterior fragments, and assayed rescue of lethality and regeneration after 30 days in vitro (*Figure 6a*). Remarkably, bulk-cell transplants were able to either partially or fully rescue irradiated worms that were destined to die (*Figure 6a,c*). 'Full' rescue was ascribed to worms with normal adult appearance whereas 'partial' rescue was assigned to cases in which proglottids were regenerated but the worms displayed abnormalities, like contracted necks (*Figure 6—figure supplement 1a*). We did not observe any proglottid regeneration in irradiated worms with or without buffer injection (*Figure 6a,c*).

Is the rescue ability described above dependent on tapeworm cycling cells? We exposed donors to F-*ara*-EdU for 1 hr, to label cycling cells prior to transplantation into irradiated hosts (*Figure 6—figure supplement 1b*). Though bulk-cell transplants were performed, injection sites contained 0, 1, or small groups of F-*ara*-EdU$^+$ cells immediately after transplantations (*Figure 6—figure supplement 1c*), likely due to technical challenges. Despite this issue, we were able to detect large colonies of F-*ara*-EdU$^+$ cells 3 days post-transplantation (*Figure 6—figure supplement 1d*). We also observed that some labeled cells were incorporated into terminally differentiated tissues at the animal edge (*Figure 6—figure supplement 1d*: inset). Thus, cycling cells from donors are able to become established and differentiate inside the irradiated host.

To test if the cycling-cell population is necessary to rescue lethally irradiated tapeworms, we depleted cycling cells from donor worms using 50 mM hydroxyurea (HU), which resulted in 96 ± 3% loss of cycling cells after 6 days (*Figure 6—figure supplement 1e–f*). Cycling cells are essential for rescue of regeneration as injected cells from HU-treated donors rescued only 1% of the time, compared to 26% rescue using cells sourced from sister donors that did not receive the drug (*Figure 6b–c*). HU was used to deplete cycling cells instead of irradiation in order to avoid inducing DNA damage in the transplanted cells. Cells transplanted from HU-treated donors had otherwise comparable morphology to untreated cells (*Figure 6—figure supplement 1g*). Our results suggest that tapeworm cycling cells contain *bona fide* stem cell activity.

With this functional assay in hand, we examined the rescue ability of cells from anterior donor tissues (including the regeneration-competent neck) compared to donor tissues from the most posterior termini of 6-day-old tapeworms (which are regeneration incompetent and exclusively comprised of proglottids). Cells from either region were able to rescue regeneration in lethally irradiated tapeworms (*Figure 6b–c*). Thus, cells from posterior proglottids were competent to receive signals from the head and neck region that regulate regenerative ability. Interestingly, using pulse-chase experiments with F-*ara*-EdU, we find that the cycling cells of posterior proglottids can give rise to multiple differentiated cell types like muscle/tegument at the animal edge as well as flame cells of the protonephridial system marked by anti-acetylated α-tubulin antibodies (*Rozario and Newmark, 2015*) (*Figure 6—figure supplement 2*). Thus, the cycling cells from tapeworm posteriors show hallmarks of stem cell activity, despite the fact that this tissue is not competent to regenerate.

Taken together, the results of our study support the idea that the regeneration competence of the neck is due to extrinsic signals that regulate regeneration, rather than intrinsic properties of stem cells in the neck region (see Discussion). It appears that in tapeworms, location matters enormously: the head and neck environment provide cues that regulate the ability of stem cells to regenerate proglottids, even though cycling cells (and likely stem cells), are not anatomically confined.

## Discussion

Across the flatworm phylum, both free-living and parasitic worms maintain stem cells throughout adulthood but display a range of regenerative abilities. The freshwater planarian *S. mediterranea* can regenerate its whole body from tiny amputated fragments (*Newmark and Sánchez Alvarado, 2002*). The blood fluke *Schistosoma mansoni* cannot regenerate after amputation, though it does employ adult somatic stem cells in other ways, such as to repair injury (*Collins and Collins, 2016*) and maintain its tegument (*Collins et al., 2016*; *Wendt et al., 2018*). Prior to this study, the regenerative ability of tapeworms had never been tested comprehensively. Although it was known that anterior fragments containing the head, neck, and immature proglottids could regenerate into fully mature tapeworms once transplanted into a rat intestine (*Read, 1967*; *Goodchild, 1958*), fragments lacking heads could not be tested for regenerative ability using transplantation. Attempts were made to suture *H. diminuta* fragments with mutilated or removed heads into a rat intestine but these fragments were invariably flushed out (*Goodchild, 1958*). Here we employ a robust in vitro culture

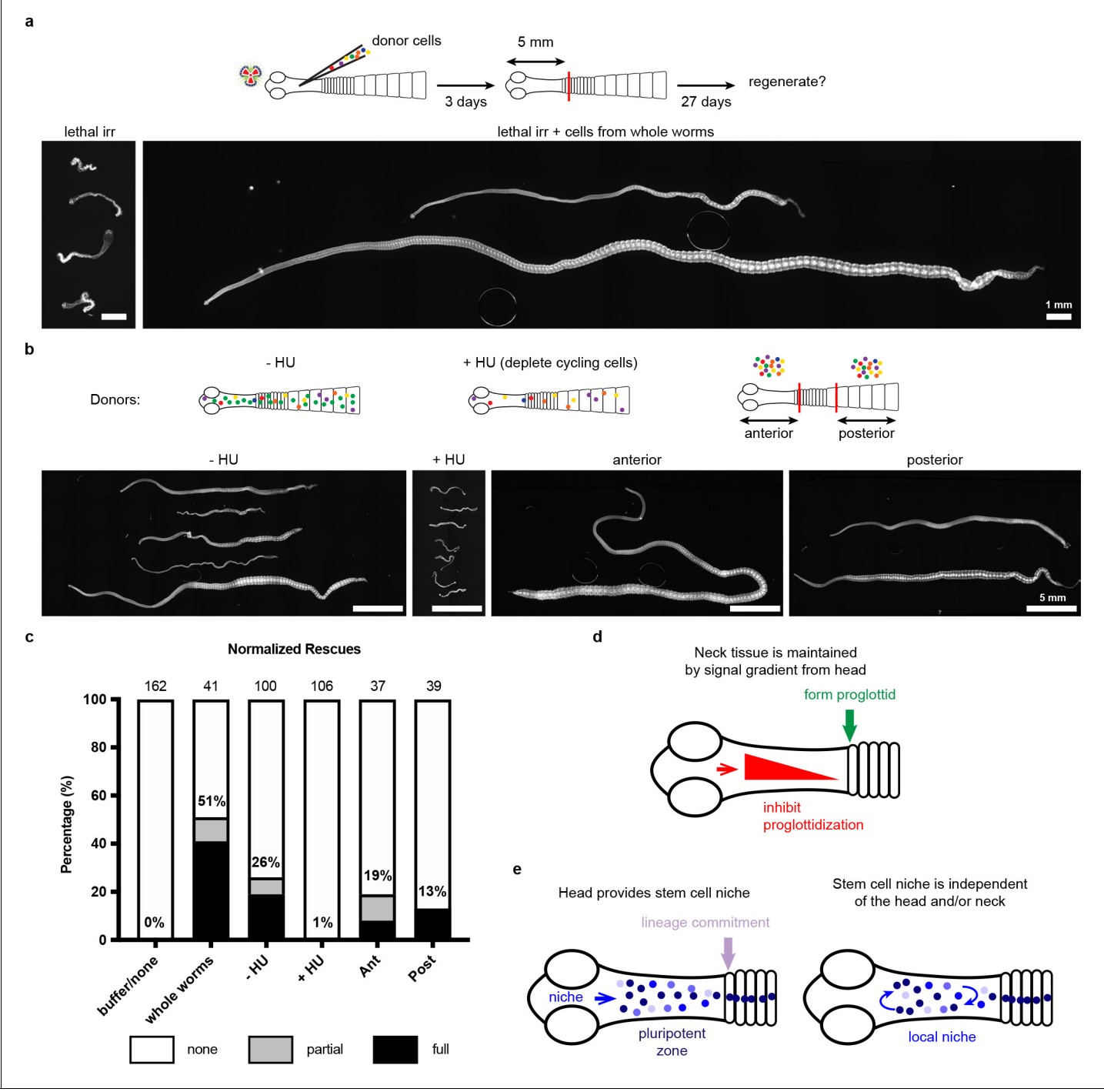

**Figure 6.** Stem cell activity depends on cycling cells but is not confined to cells from the neck. (**a-b**) DAPI-stained worms after rescue with cell transplantations from whole-worm donors (**a**) or sourced from depicted donors (**b**). (**c**) Quantification of rescue phenotypes from pooled experiments. Number of animals listed above bars. (**d**) Model for head-dependent neck maintenance and proglottid formation. (**e**) Models of head-dependent or -independent stem cell niches.

DOI: https://doi.org/10.7554/eLife.48958.017

The following figure supplements are available for figure 6:

**Figure supplement 1.** Stem cell activity depends on cycling cells.
DOI: https://doi.org/10.7554/eLife.48958.018

**Figure supplement 2.** Cycling cells give rise to multiple lineages in both anterior and posterior fragments.
DOI: https://doi.org/10.7554/eLife.48958.019

system that allowed us to test regeneration of any amputated *H. diminuta* fragment for the first time. We show that the neck is both necessary and sufficient for proglottid regeneration, though this regenerative ability is ultimately finite without regulatory signals that depend on the presence of the head. *H. diminuta* is an intriguing model to discover signals that both drive and limit regenerative ability.

During homeostasis, the neck of *H. diminuta* serves as a 'growth zone' from which proglottids are thought to bud one at a time (*Lumsden and Specian, 1980*), thus, it makes intuitive sense that this tissue would retain the ability to regenerate proglottids post-amputation. Furthermore, cells with the typical morphology of stem cells reside in the neck (*Bolla and Roberts, 1971*). However, we find that cycling cells are present in all regions regardless of regenerative competence. Thus, it was necessary to embark on a more thorough characterization of tapeworm cycling cells to understand how *H. diminuta* may regulate stem cells and enable proglottid regeneration.

We depleted cycling cells in *H. diminuta* using irradiation and employed RNA-seq to uncover potential stem cell regulators in an unbiased fashion. Though irradiation may have secondary effects beyond stem cell depletion (*Solana et al., 2012*), this approach allowed us to generate an initial list of candidate tapeworm stem cell genes. Using dFISH, we were able to verify 38 transcripts that were expressed at least partially in cycling cells, providing the first molecular characterization of this population in *H. diminuta*.

Adult somatic stem cells in free-living flatworms have already been well described molecularly, and share many conserved regulators. However, parasitic flatworms have lost some stem cell genes (e.g. *piwi*, *vasa*, and *tudor*) (*Tsai et al., 2013*) but retained others (e.g. *argonaute*, *fgfr*) (*Collins et al., 2013*; *Koziol et al., 2014*). How do the cycling-cell transcripts we identified in *H. diminuta* compare to stem cells in free-living planarians? (*Fincher et al., 2018*; *Plass et al., 2018*; *Labbé et al., 2012*; *Rozanski et al., 2019*) (*Supplementary file 1c*). Of 38 verified tapeworm cycling-cell transcripts, 28 had putative planarian homologs (tblasx E-value $<10^{-10}$) though not all were reciprocal blast hits. 16 of these planarian transcripts were designated as cluster-defining genes in the Fincher et al. single-cell sequencing dataset and 6/16 are neoblast cluster-defining genes. Plass et al. also performed single-cell sequencing of planarians but most of the putative planarian homologs of tapeworm cycling-cell transcripts that we identified were not found in their dataset. However, 8/28 transcripts showed enriched expression in neoblast clusters. We also compared the expression of these planarian transcripts in RNA-seq of three cell populations sorted by fluorescence-activated cell sorting (FACS): 1) X1 (neoblasts in G2/M), 2) X2 (G1 neoblasts and progenitors), and 3) Xins (differentiated cells) (*Labbé et al., 2012*; *Rozanski et al., 2019*). We find that 22/28 putative planarian homologs of tapeworm cycling-cell transcripts show enriched expression in either the X1 or X2 populations, which contain neoblasts. Thus, despite >500 million years of separation between free-living and parasitic flatworm evolution (*Laumer et al., 2015*), tapeworm cycling-cell transcripts have conserved signatures with planarian neoblasts.

In multiple species of flatworms, stem cells have been shown to be transcriptionally heterogenous (*Solana et al., 2012*; *Fincher et al., 2018*; *van Wolfswinkel et al., 2014*; *Zeng et al., 2018*). For example, in larvae of the tapeworm *Echinococcus multilocularis*, many putative stem cell markers show limited overlapping gene expression patterns (*Koziol et al., 2014*). In keeping with these findings, we observe transcriptional heterogeneity within the cycling-cell population of *H. diminuta*. Additionally, we identified 23 transcripts, including *zmym3* and *pogzl*, that label all cycling cells. Importantly, we were able to use RNAi to functionally verify that cycling-cell genes like *zmym3* and *pogzl* are critical for stem cell maintenance and that inhibition of these genes leads to impaired growth and regeneration. Both *zmym3* and *pogzl* are neoblast cluster-defining genes in planarians (*Supplementary file 1c*) suggesting that their functions in stem cell regulation may be conserved across the two species. In fact, the planarian homolog of tapeworm *pogzl*, *factor initiating regeneration 1* (*fir1*), was recently shown to be expressed in planarian neoblasts (*Han, 2018*). RNAi of *fir1* resulted in decreased cell division after amputation and failure to regenerate blastemas (*Han, 2018*). On the other hand, the function of *zmym3* in regeneration is not known, but in other systems, *zmym3* regulates cell cycle progression (*Hu et al., 2017*) and DNA repair (*Leung et al., 2017*), two essential functions for stem cells. Coincidentally, both *zmym3* and *pogzl* are Zn finger proteins with predicted DNA-binding activity and could function as transcriptional regulators of stem cells. Thus, it would be interesting to further understand the mechanism of action of *zmym3* and *pogzl* in stem cells of parasitic and free-living flatworms.

In this study, we showed the first use of RNAi in *H. diminuta*. RNAi has been demonstrated previously in other tapeworm species (*Pierson et al., 2010*; *Mizukami et al., 2010*; *Spiliotis et al., 2010*), though it has not been widely adopted for studying tapeworm biology due to technical challenges like poor knockdown efficacy, inefficient penetrance, and the difficulty of in vitro culture. Taking advantage of the robust in vitro culture of *H. diminuta*, our RNAi scheme can be expanded to ascertain functions for many parasitic flatworm genes that thus far have been refractory to functional analyses.

Our screening strategy allowed us to verify genes with enriched expression in some or all cycling cells; however, none of these genes were expressed exclusively in the neck. Since we had observed that regenerative ability was anteriorly biased across the neck, we attempted to leverage this observation and query whether a subpopulation of pluripotent cycling cells may be asymmetrically distributed across the neck and would be identifiable by RNA-seq. Through A-P transcriptional profiling of the neck, we identified 461 anterior-enriched transcripts, but the vast majority of them were neither irradiation-sensitive nor detected in cycling cells by dFISH. Thus, a subpopulation of neck-resident pluripotent stem cells, seems unlikely to explain the region-specific regenerative ability of tapeworms. Nonetheless, our study does not exclude the existence of a subpopulation of pluripotent stem cells that may be stably maintained in the adult. Future studies using single-cell RNA sequencing are likely to provide a thorough characterization of adult somatic stem cells in *H. diminuta*, as has been the case for planarians (*Fincher et al., 2018*; *Plass et al., 2018*; *Zeng et al., 2018*).

Is the neck competent to regenerate because of a unique stem cell population that has yet to be identified, or because of signals extrinsic to stem cells that make the neck permissive for regeneration? Tapeworms exposed to a lethal dose of irradiation prior to amputation are not competent to regenerate and will eventually degenerate and die. However, transplantation of cells from wild-type donors into the necks of irradiated tapeworms rescued lethality and regeneration. This rescue ability is severely compromised if donor worms are first depleted of cycling cells using drug treatment with HU, suggesting that some or all cycling cells have stem cell ability. Interestingly, stem cell ability is not restricted to cells from regeneration-competent regions: cells from posterior tissues that do not regenerate proglottids are still able to rescue regeneration when transplanted into the neck. These data strongly suggest that the microenvironment within the neck confers regenerative ability to this region.

The interplay between intrinsic and extrinsic stem cell regulatory signals has been shown to play important roles in regeneration. Head regeneration was induced in three naturally regeneration-deficient planarian species by manipulating the gradient of Wnt signaling by RNAi (*Sikes and Newmark, 2013*; *Liu et al., 2013*; *Umesono et al., 2013*). These planarians maintain pluripotent stem cells but do not normally regenerate heads from posterior tissues due to inappropriately high levels of Wnt signaling, which inhibit anterior regeneration. As in planarians, gradients of Wnt signaling delineate A-P polarity in tapeworms (*Koziol et al., 2016*). Our transcriptional profiling of the neck A-P axis has already revealed hundreds of candidate genes with polarized expression profiles. Future experiments will help clarify how Wnt signaling and other A-P axis regulation in the neck impacts tapeworm regeneration.

Several plausible models can explain region-specific regeneration in *H. diminuta*. Head-dependent signals may create gradients across the neck that inhibit proglottidization and are necessary to maintain the neck as an unsegmented tissue. Proglottids can only form once the inhibitory signals are sufficiently diminished (*Figure 6d*). In this model, the neck is competent to regenerate because of its juxtaposition to the head. After decapitation, the head-dependent signals eventually dissipate and segmentation signals dominate at the expense of the neck. The cellular source of the head-dependent signals and their molecular identity will be exciting avenues for future research.

In addition to its function in maintaining the neck, the head may also play a role in stem cell regulation (*Figure 6e*). The head may regulate a niche (directly or indirectly) that is necessary for the maintenance of pluripotency in the neck. In this model, stem cells are collectively pluripotent only when they receive head-dependent niche signals, thus limiting regenerative potential to the neck. Alternatively, stem cells may depend on a local niche that is independent of the head. In this model, stem cells have the capacity to form all cell lineages from any amputated fragment; however, the extrinsic signals that activate proglottid formation are restricted to the posterior neck region. Identifying the stem cell niche and its relationship to the head and neck microenvironment will provide crucial insights into our understanding of tapeworm regeneration.

## Conclusion

Our study shows that *H. diminuta* is a powerful developmental model for understanding intrinsic and extrinsic regulation of stem cells and regeneration. The regionally limited regenerative biology of *H. diminuta* and the technical advances put forth in this work show that we can exploit this tapeworm to understand the complexities of stem cell regulation in parasites. We defined heterogenous stem cells that are collectively pluripotent but that require extrinsic head-dependent signals to enable persistent proglottid regeneration. Understanding how the stem cell niche we describe is regulated may have broad implications for elucidating stem cell biology in parasitic flatworms, as well as other regenerative animals.

# Materials and methods

**Key resources table**

| Reagent type (species) or resource | Designation | Source or reference | Identifiers | Additional information |
|---|---|---|---|---|
| Strain, (*Hymenolepis diminuta*) | BioSample accession SAMN11958994 | Carolina Biologicals | Cat# 132232 | |
| Antibody | anti-Oregon Green 488-HRP antibody (rabbit polyclonal) | Invitrogen | A21253 | IF(1:1000) |
| Antibody | anti-DIG-AP (sheep polyclonal) | Sigma Aldrich | Cat# 11093274910 | IF(1:2000) |
| Antibody | anti-DIG-POD (sheep polyclonal) | Sigma Aldrich | Cat#: 11207733910 | IF(1:2000) |
| Antibody | anti-DNP-HRP (rabbit polyclonal) | Vector Laboratories | Cat#: MB-0603 | IF(1:2000) |
| Antibody | anti-acetylated α-tubulin (mouse monoclonal) | Santa Cruz | Cat#: sc-23950 | IF(1:500) |
| Sequence-based reagent | PCR primers | This paper | | *Supplementary file 1E* |
| Sequence-based reagent | Transcriptome Shotgun Assembly (*Hymenolepis diminuta*) | DDB/ENA/Genbank | GHNR01000000 | |
| Sequence-based reagent | Sequence Read Archives for transcriptome assembly | DDB/ENA/Genbank | PRJNA546290 | SRX6045715-SRX6045719 |
| Sequence-based reagent | Sequence Read Archives for differential gene expression | DDB/ENA/Genbank | PRJNA546293 | SRX6064929-SRX6064933 |
| Recombinant DNA reagent | Plasmid- pJC53.2 | Addgene | 26536 | |
| Chemical compound, drug | F-*ara*-EdU | Sigma Aldrich | T511293 | 0.1 µM (in 1% final DMSO concentration) |
| Chemical compound, drug | Oregon green 488-azide | Invitrogen | O10180 | 100 µM |
| Chemical compound, drug | Hydroxyurea | Sigma Aldrich | Cat#: H8627 | 50 mM |

## Animal care and use

Infective *H. diminuta* cysts were obtained from Carolina Biological (132232). To obtain adult tapeworms, 100–400 cysts were fed to Sprague-Dawley rats by oral gavage in ~0.5 mL of 0.85% NaCl. Rats were euthanized in a $CO_2$ chamber 6 days post-gavage, tapeworms were flushed out of the small intestine, and washed in 1X Hanks Balanced Salt Solution (HBSS; Corning) (140 mg/L $CaCl_2$, 100 mg/L $MgCl_2.6H_2O$, 100 mg/L $MgSO_4.7H_2O$, 400 mg/L KCl, 60 mg/L $KH_2PO_4$, 350 mg/L

NaHCO$_3$, 8 g/L NaCl, 48 mg/L Na$_2$HPO$_4$, 1 g/L D-glucose, no phenol red). Rodent care was in accordance with protocols approved by the Institutional Animal Care and Use Committee (IACUC) of the University of Wisconsin-Madison (M005573).

## In vitro parasite culture

Biphasic parasite cultures were prepared based on the Schiller method (Schiller, 1965). Briefly, the solid phase was made in 50 mL Erlenmeyer flasks by mixing 30% heat-inactivated defibrinated sheep blood (Hemostat) with 70% agar base for 10 mL blood-agar mixture per flask. Fresh blood was heat-inactivated at 56°C for 30 min then kept at 4°C and used repeatedly for one week by first warming the blood to 37°C. The agar base was prepared from 8 g Difco nutrient agar and 1.75 g NaCl in 350 mL water, autoclaved, and stored at 4°C. Before use, the agar base was microwaved to liquify, and cooled to below 56°C before mixing with warmed blood. After the blood-agar mixture solidified, 10 mL of Working Hanks 4 (WH4; 1X HBSS/4 g/L total glucose/1X antibiotic-antimycotic (Sigma)) was added. Each flask was topped with a gas-permeable stopper (Jaece Identi-plug) and pre-incubated at 37°C in hypoxia (3% CO$_2$/5% O$_2$/92% N$_2$) overnight before use. Before tapeworms were transferred into the flasks, the liquid phase was adjusted to pH7.5 with 200 µL 7.5% NaHCO$_3$ (Corning). Tapeworms were first washed in WH4 for 10 mins at 37°C in petri dishes pre-coated with 0.5% BSA to inhibit sticking. Transfers to pre-cultured flasks were performed by gently lifting the worms with a stainless-steel hook (Moody Tools) and immersing them in the liquid phase. Tapeworms were grown in hypoxia and transferred to fresh cultures every 3–4 days.

## Fixation and DAPI staining

Tapeworms were heat-killed by swirling in 75°C water for a few seconds until the worms relaxed and elongated, then fixative (4% formaldehyde in Phosphate Buffered Saline with 0.3% TritonX-100 (PBSTx)) was added immediately for 30 min-2hr at room temperature or overnight at 4°C. For DAPI staining, samples were incubated in 1 µg/mL DAPI (Sigma) in PBSTx overnight at 4°C and cleared in 80% glycerol/10 mM Tris pH7.5/1 mM EDTA overnight at room temperature before mounting.

## F-ara-EdU uptake and staining

For F-ara-EdU pulse, tapeworms were incubated in 0.1 µM F-ara-EdU (Sigma) in 1% DMSO at 37°C in WH4. Tapeworms were heat-killed (above) and fixed in 4% formaldehyde/10% DMSO/1% NP40/PBSTx. Large tissues/worms were permeabilized by incubating in PBSTx at room temp for several days. Additional permeabilization was achieved by treatment with 10 µg/mL Proteinase-K/0.1% SDS/PBSTx for 10–30 min at room temperature, fixed in 4% formaldehyde/PBSTx for 10 min before samples were cut into small pieces or retained whole in PBSTx. Samples were further permeabilized in PBSTx/10% DMSO/1% NP40 for 20 min-1 hr (depending on size) before performing the click-it reaction (Salic and Mitchison, 2008) with Oregon Green 488 azide (Invitrogen). Signal was detected using anti-Oregon Green 488-HRP antibody (1:1000; Invitrogen) in K-block (5% Horse serum/0.45% fish gelatin/0.3% Triton-X/0.05% Tween-20/PBS) (Collins et al., 2011) followed by 10–20 min Tyramide Signal Amplification (TSA) reaction (King and Newmark, 2013). Tiled confocal z-stacks through the anterior of the worms were taken and cell numbers were counted using background subtraction on Imaris software. F-ara-EdU$^+$ cells were normalized to worm area from maximum projections of the DAPI stain. Flame cells were stained using an anti-acetylated α-tubulin mouse antibody at 1:500 (sc-23950, Santa Cruz) as described previously (Rozario and Newmark, 2015).

## Irradiation

Most irradiation was performed using a CellRad irradiator (Faxitron Bioptics) at 200 Gy (150 kV, 5 mA) with two exceptions. Due to instrument failure, a cesium irradiator was used for one rescue experiment with donors + /- HU (Figure 6b) at 400 Gy (92 ± 5% cycling cell loss 3 days post-irradiation). The rescue experiment with + /- HU donors was performed a third time once we gained access to an x-irradiator (Xstrahl RS225 Cell Irradiator), where the lethal dose was 200 Gy (63 ± 10% cycling cell loss 3 days post-irradiation). All three experiments gave similar results despite the use of different irradiators. In all cases, lethal irradiation was determined as the dosage at which tapeworms degenerated, had 0 proglottids, and were inviable after 30 days in culture. Irradiation was performed in WH4 in BSA-coated petri dishes.

## Transcriptome assembly

RNA was collected from five regions: 1) head and neck, 2) immature proglottids, 3) mature reproductive proglottids, 4) gravid proglottids, and 5) mixed larval stages isolated from beetles. The first three regions covered the entirety of 3.5-week-old adult tapeworms. Gravid proglottids were taken from posteriors of 10-week-old tapeworms. Paired-end libraries were constructed with $2 \times 150$ bp reads from a HiSeq2500 chip. $2 \times \sim 30$ million reads were obtained for each sample. The transcriptome was assembled from three components: 1) map-based assembly, 2) de novo assembly, and 3) Maker predictions from Wormbase Parasite. The map-based assembly was performed using TopHat2 with the 2014 *H. diminuta* draft genome courtesy of Matt Berriman (Wellcome Sanger Institute, UK). 15,859 transcripts were assembled using TopHat. De novo assembly was performed using Velvet/Oases and resulted in 144,682 transcripts. There were 11,275 predicted Maker transcripts and 73.2% matched (>95% along the length) to the TopHat transcripts. The remaining predicted transcripts that were not represented in the TopHat dataset were added for a combined TopHat/predicted set of 17,651 transcripts. Most of the Oases transcripts matched to the TopHat/predicted set but 35,300 or 24.4% of the Oases transcripts did not (>75% match cut-off). These transcripts could be transcripts missed in the genome, transcription noise, non-coding transcripts, or contamination. We found significant contamination from beetle tissue in the larval tapeworm sample (more below). Initial filtering for contamination excluded 1388 transcripts (from beetle, rat, bacterial, and viral sources). At this point 51,563 transcripts were retained from the three methodologies described above and were processed for further filtering.

There was significant contamination from beetle tissues that had adhered to the tapeworm larvae, which produced transcripts with best hits to beetle proteins (*Ixodes scapularis, Harpegnathos saltator, Monodelphis domestica, Nasonia vitripennis, Pediculus humanus corporis, Solenopsis invicta, Tenebrio molitor,* or *Tribolium castaneum*). Most of the transcripts were from the Oases de novo assembly and did not match the *H. diminuta* genome. Furthermore, they were strongly over-represented in the larval sample only. To filter out beetle contamination, we removed 11,918 transcripts from the Oases assembly without matches to the *H. diminuta* genome that showed >90% expression (by RPKM) in the larval sample.

To the remaining 39,645 transcripts, we applied additional filters: 1) Remove transcripts if average RPKM <1 unless the transcript is long (>1000 bp), has a long ORF (>500 bp) or is annotated. 11,615 transcripts were removed as they met none of these criteria. 2) A stringent expression cut-off was applied to the remaining Oases transcripts; transcripts were discarded if average RPKM <5 and maximum RPKM <10 unless the transcripts were long (>1000 bp), had long ORFs (>500 bp) or were annotated. 8027 transcripts were removed. 3) 51 transcripts were removed because they are mitochondrial or rRNAs. 4) An ORF size filter was applied to remove all transcripts with ORF <300 bp unless they are annotated. 5331 transcripts were removed. 5) For the Maker predicted transcripts, expression and size filters were applied to remove transcripts with expression <1 RPKM and size <500 bp. 275 transcripts were removed.

Our final transcriptome is comprised of 14,346 transcripts (84.9% TopHat, 8.4% Maker predictions, 6.1% Oases with match to genome, and 0.6% Oases without match to genome). The total transcriptome size is 34 Mb with average transcript length of 2,354 bp. This Transcriptome Shotgun Assembly project has been deposited at DDB/ENA/Genbank under the accession GHNR00000000. The version described in this paper is the first version, GHNR01000000. All sequence reads are available at GenBank Bioproject PRJNA546290.

## RNA-seq for differential gene expression analyses

Tissue was collected and immediately frozen on dry ice in 100 µL Trizol (Life Technologies) before RNA extraction. Tissue homogenization was performed as the mixture was in a semi-frozen state using RNase-free pestles and a pestle motor. RNA was purified using the Direct-zol RNA MiniPrep kit (Zymo). RNA quality was assessed using Bioanalyzer, libraries were prepared with TruSeq Stranded mRNAseq Sample Prep kit (Illumina), and sequenced on two lanes on a HiSeq2500 chip. We performed paired-end sequencing and obtained ~20 million reads per sample. Samples were obtained in triplicate. To identify irradiation-sensitive transcripts, 2 mm anterior tapeworm fragments were cut from 10 worms after 3 days in vitro. To identify differentially expressed transcripts across the neck A-P axis, 1 mm fragments were cut from 20 freshly obtained 6-day-old tapeworms. Paired-

end reads were mapped to the transcriptome (above) using default settings on CLC Genomics Workbench 6 (Qiagen) except that read alignments were done with a relaxed length fraction of 0.5. Differential gene expression analysis was done with the same software using estimate tagwise dispersions on total read counts and a total count filter cut-off of 5 reads. All sequence reads used for differential gene expression analyses are available at GenBank Bioproject PRJNA546293.

## Cloning

Target genes were amplified using PCR with Platinum Taq (Life Technologies) from cDNA generated from RNAs extracted from tapeworm anteriors to enrich for neck transcripts. PCR products were inserted via TA-mediated cloning into the previously described vector pJC53.2 (*Collins et al., 2010*) pre-digested with *Eam11051*. Anti-sense riboprobes could be generated by in vitro transcription with SP6 or T3 RNA polymerases. For RNAi, dsRNA was generated using T7 RNA polymerase. For sequences and primers, refer to *Supplementary file 1e*.

## In situ hybridization

WISH and FISH protocols were modified from previously published methods for planarians (*King and Newmark, 2013*) and the mouse bile-duct tapeworm *Hymenolepis microstoma* (*Olson et al., 2018*). Tapeworms were heat killed and fixed in 4% formaldehyde/10% DMSO/1% NP40/PBSTx for 30 min at room temperature before washing and dehydration into methanol. Dehydrated samples were frozen at −30°C for at least 2 days. After rehydration, samples were permeabilized in 10 µg/mL Proteinase-K/0.1% SDS/PBSTx for 30 min, washed into 0.1 M Triethanolamine pH7-8 (TEA), 2.5 µL/mL acetic anhydride was added for 5 min with vigorous swirling, acetic anhydride step was repeated, washed in PBSTx, and post-fixed in 4% formaldehyde/PBSTx for 10 min. Probe synthesis, hybridization, and staining were performed as previously described (*King and Newmark, 2013*) using probe concentrations at ~50 ng/mL for 16–48 hr at 56°C. All probes were synthesized with either DIG or DNP haptens and detected using the following antibodies, all at 1:2000: anti-DIG-AP (Sigma), anti-DIG-POD (Sigma), anti-DNP-HRP (Vector Labs). Colorimetric development was done using NBT (Roche)/BCIP (Sigma) or with Fast-Blue (Sigma) (*Currie et al., 2016*). Fluorescent signal was visualized after 10–20 min TSA reaction (*King and Newmark, 2013*). DAPI staining and mounting were performed as described above.

## Imaging

Confocal imaging was performed on a Zeiss LSM 880 with the following objectives: 20X/0.8 NA Plan-APOCHROMAT, 40X/1.3 NA Plan-APOCHROMAT, and 63X/1.4 NA Plan-APOCHROMAT. WISH samples and whole-mount DAPI-stained worms were imaged using Zeiss AxioZoom V16 macroscope. Image processing was performed using ImageJ for general brightness/contrast adjustments, maximum-intensity projections, and tile stitching (*Preibisch et al., 2009*).

## RNAi

dsRNA was synthesized as previously described (*Rouhana et al., 2013*) and resuspended at concentrations ~ 1.5–2 µg/µL. For control injections, 1.5 kb dsRNA derived from *ccdB* and *camR*-containing insert of the pJC53.2 vector was used (*Collins et al., 2010*). 6-day-old tapeworms were obtained and microinjected with dsRNA using femtotips II via the Femtojet injection system (Eppendorf) to obtain spreading across the first ~3–4 mm anterior of the tapeworm. The spread of injected fluids could be detected by a temporary increase in opacity. 500 hPa injection pressure for 0.3–1 s was used per injection site. Whole tapeworms were cultured in vitro for 3 days, 2 mm anterior fragments were amputated, worms were re-injected with dsRNA on day 6, and cultured in vitro for an additional 9 days before termination.

## qPCR for target gene knockdown efficacy

Whole worms (6 days old) were injected with dsRNA throughout and frozen in Trizol on dry ice after 6 days in vitro for RNA extraction according to manufacturer's protocol and DNAse (Promega) treatment for 30 min at 37°C. cDNA synthesis was performed using SuperScriptIII First-Strand Synthesis System (Invitrogen) with Oligo(dT)$_{20}$ primers followed by iScript cDNA Synthesis Kit (Bio-Rad). qPCR was performed using GoTaq Mastermix (Promega) on a StepOnePlus real-time PCR machine

(Applied Biosystems). *60S ribosomal protein L13* (*60Srpl13*) was used as an internal normalization control. For primers refer to *Supplementary file 1e*.

## Hydroxyurea (HU) treatment

Tapeworms were treated with HU (Sigma) or HBSS (for controls) every day for a total of 6 days. HU stock solution was made fresh every day at 2 M in HBSS. 250 μL was added to each flask of tapeworms for final concentration of 50 mM. HU is unstable at 37°C so worms were transferred into fresh HU-containing media every two days, and fresh HU was added every other day.

## Cell transplantations

For dissociated cell preparations, tapeworms were placed in a drop of calcium-magnesium free HBSS (CMF HBSS, Gibco), minced into small pieces with a tungsten needle, incubated in 3X Trypsin-EDTA (Sigma) in CMF HBSS for 30 min at 37°C and dissociated using a dounce homogenizer (Kontes). Cells were pelleted by centrifugation at 250 g for 5 min. The cell pellet was washed in CMF HBSS and passed through cell strainers at 100 μm, 40 μm, 20 μm, and 10 μm (Falcon and Sysmex) with one spin and wash in between. Cells were pelleted and resuspended in 200–400 μL WH4 with 0.05% BSA. Cell injections were performed using the Cell Tram Oil four injection system (Eppendorf) into the necks of irradiated worms. For + /- HU donors, cell concentrations were measured using a hemocytometer and normalized (to ~$10^8$ cells/mL) to ensure equal numbers of cells were injected. For all rescue experiments, cells were injected into irradiated hosts on the same day that the hosts were irradiated. After 3 days in vitro, 5 mm anterior fragments were amputated and grown for an additional 27 days.

## Statistical analysis

Statistical analyses were performed using Prism7 software (GraphPad Prism). All experiments were repeated at least twice. All measurements were taken from distinct samples. Error bars, statistical tests, number of replicates (N) and sample sizes (n) are indicated in corresponding figure legends. Either Dunnett's or Tukey's multiple comparison tests were used for one-way ANOVAs. SD = standard deviation. P-values: ns = not significant, *=$p \leq 0.5$, ****=$p \leq 0.0001$.

# Acknowledgements

We thank: members of the Newmark lab, especially Melanie Issigonis and Umair Khan, for discussions and comments on the manuscript; Alvaro Hernandez (Roy J Carver Biotechnology Center, University of Illinois at Urbana-Champaign) for RNA-seq; and Bret Duffin (Morgridge Institute for Research) for invaluable assistance with irradiation. This work was supported by NIH R21AI119960. PAN is an investigator of the Howard Hughes Medical Institute.

# Additional information

### Competing interests

Phillip A Newmark: Reviewing editor, *eLife*. The other authors declare that no competing interests exist.

### Funding

| Funder | Grant reference number | Author |
| --- | --- | --- |
| National Institute of Allergy and Infectious Diseases | R21AI119960 | Phillip A Newmark |
| Howard Hughes Medical Institute | | Phillip A Newmark |

The funders had no role in study design, data collection and interpretation, or the decision to submit the work for publication.

## Author contributions
Tania Rozario, Conceptualization, Supervision, Investigation, Methodology, Writing—original draft, Project administration, Writing—review and editing; Edward B Quinn, Jianbin Wang, Investigation, Writing—review and editing; Richard E Davis, Supervision, Methodology, Writing—review and editing; Phillip A Newmark, Conceptualization, Supervision, Funding acquisition, Writing—original draft, Project administration, Writing—review and editing

## Author ORCIDs
Tania Rozario https://orcid.org/0000-0002-9971-5211
Jianbin Wang https://orcid.org/0000-0003-3155-894X
Phillip A Newmark https://orcid.org/0000-0003-0793-022X

## Ethics
Animal experimentation: Rodent care was in accordance with protocols approved by the Institutional Animal Care and Use Committee (IACUC) of the University of Wisconsin-Madison (M005573).

## Decision letter and Author response
Decision letter https://doi.org/10.7554/eLife.48958.030
Author response https://doi.org/10.7554/eLife.48958.031

# Additional files

## Supplementary files
• Supplementary file 1. *Supplementary file 1a* Irradiation-sensitive transcripts identified by RNA-seq. *Supplementary file 1b* Summary of dFISH experiments with irradiation-sensitive transcripts and cycling cell markers *h2b* and/or *mcm2*. *Supplementary file 1c* Comparative analysis of verified tapeworm cycling-cell transcripts to gene expression datasets for planarian neoblasts. *Supplementary file 1d* Anterior-enriched and anterior-depleted neck transcripts by RNA-seq. *Supplementary file 1e* Sequences and primers for all genes reported.
DOI: https://doi.org/10.7554/eLife.48958.020

• Transparent reporting form DOI: https://doi.org/10.7554/eLife.48958.021

• Reporting standard 1. MINSEQE: Minimum Information about a high-throughput Nucleotide SeQuencing Experiment - a proposal for standards in functional genomic data reporting.
DOI:

## Data availability
Sequencing data have been deposited in DDB/ENA/Genbank under accession codes GHNR01000000, PRJNA546290 and PRJNA546293.

The following datasets were generated:

| Author(s) | Year | Dataset title | Dataset URL | Database and Identifier |
|---|---|---|---|---|
| Rozario T, Quinn EB, Wang J, Davis RE, Newmark PA | 2019 | Hymenolepis diminuta transcriptome | http://www.ncbi.nlm.nih.gov/bioproject/546290 | BioProject, PRJNA546290 |
| Rozario T, Quinn EB, Wang J, Davis RE, Newmark PA | 2019 | Region-specific regulation of stem cell-driven regeneration in tapeworms | https://www.ncbi.nlm.nih.gov/bioproject/546293 | BioProject, PRJNA546293 |
| Rozario T, Quinn EB, Wang J, Davis RE, Newmark PA | 2019 | Hymenolepis diminuta transcriptome shotgun assembly | https://www.ncbi.nlm.nih.gov/nuccore/GHNR00000000.1/ | NCBI, GHNR01000000 |

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
