## [Decision Letter]

Thank you for submitting your article "Region-specific regulation of stem cell-driven regeneration in tapeworms" for consideration by *eLife*. Your article has been reviewed by three peer reviewers, including Yukiko M Yamashita as the Reviewing Editor and Reviewer #1, and the evaluation has been overseen by Marianne Bronner as the Senior Editor. The following individual involved in review of your submission has agreed to reveal their identity: Peter W Reddien (Reviewer #3).

The reviewers have discussed the reviews with one another and the Reviewing Editor has drafted this decision to help you prepare a revised submission.

This work is the first comprehensive study to analyze regeneration of tapeworm *H. diminuta* at a molecular level and establishes it as a model system to study striking regeneration capacity. Accordingly, all reviewers agreed that this study represents an important advancement in the field of regenerative biology. Essentially all comments raised by reviewers are technical ones and are straightforward to address.

Therefore, we would like to invite you to submit a revised version that addresses the reviewers' comments below. We look forward to receiving the revised manuscript.

Reviewer #1:

This work establishes tapeworm (*H. diminuta*) as a model system for molecular studies on stem cell-based tissue homeostasis. Given their capacity to live for extremely long (if not infinite), understanding the underlying mechanisms of their regeneration will be of significant interest and importance. The most important conclusion from this paper is that regenerative capacity is confined within neck, but it requires head (likely as the 'niche') to maintain long-term regeneration. Head is only the niche, but does not have stem cells on its own. And this niche appears to play a dominant role as all regions (including body) appear to contain 'proliferative' cells.

Experiments are conducted to a highest standard, and this study represents a major advance in the field by establishing tapeworm as a stem cell model system. My comments are mostly about writing: I felt that the writing/data presentation can be improved such that general readers (who are interested in stem cells but are not familiar with tapeworm anatomy – which are the majority of readers).

Specific comments:

- The fourth paragraph of the Results states that the only proliferative somatic cells are undifferentiated in tapeworm, citing Bolla and Roberts, 1971 and Sulgostowska, 1972. But these references are from the 70s, and I am not sure (or more in general, readers will wonder) what kind of methods in the 70s could provide such a conclusion at the accuracy of today's standard. Also, if old studies are that conclusive, this study would sound like not providing much of new insights. Maybe a bit more of elaboration (what was done before, and how it compares to the present study, in providing more accurate information) would be helpful.

- Figure 1B: the lack of regeneration with 'body only' is not obvious due to different scales used between day 0 vs. day 9.

- Figure 2 used irradiation to deplete proliferating cells, which the authors assumed to eliminate stem cells. Whereas I agree that this method has been successfully used to identify neoblast-specific genes in planaria, the formal possibility remains that irradiation induce transient quiescence of stem cells. If so, this experiment will be only identifying 'proliferation-associated' genes, instead of 'stem cell specific genes'. In case of planaria, the use of 'lethal dose' (i.e. no stem cells are left indeed) excluded such a possibility. I am guessing 200 Gy for tapeworm is the same treatment, but it is not explained in the paragraph where this experiment is explained (Results, fifth paragraph). Clarification might help here.

- Figure 3 explains proliferating cells are in neck parenchyma. Abstract/Introduction primed me/readers to think that stem cells are not limited to neck region and are present everywhere. However, in this figure, the authors' description focuses on neck area (starting in the sixth paragraph of the Results, they provide detailed location of cycling cells within the neck region, without mentioning the other area of the worm-like body), which inevitably made me wonder whether other regions also exhibit similar expression patterns or not, and I was quite confused going through the explanation around here. I wonder these questions might not occur to authors, because the images may be self-explanatory to them. But those who are not familiar to tapeworm anatomy cannot quite tell where the neck is/where the body starts in the presented images (labelling those regions in the panels will greatly help).

- Results, ninth paragraph: Reduction in *mcm2* is used to conclude that cell populations are gone after RNAi of *h2b* etc. However, as *mcm2* itself is a cell proliferation gene, it is still possible that the cell population still exists while reducing the expression of *mcm2*. Here I am not asking to do additional experiments to distinguish these possibilities (cell population is gone vs. proliferation is diminished). I am simply pointing out a flaw in interpretation so that they can adjust their statement.

- Figure 5-6 conclude that stem cells are not limited to the neck region. This is based on the lack of any transcripts that are differentially expressed between posterior vs. anterior regions, and the fact that cells from any regions can rescue lethally irradiated animals. Based on these data, the authors propose that head/neck serves to provide extrinsic signals to maintain stem cells, yet there are no intrinsic differences among stem cells. They also nicely show that cycling cells contain the stem cell population (by HU-induced depletion of cycling cells). Whereas the data are striking and clear, the explanation seems to be somewhat confusing (or indicating something is missing). ---upon reading the Discussion, I see that most of the issues (below) are indeed discussed well, but as I read through the result section, the description went on without addressing some major question. It might be helpful to slip in a few sentences also in Result section to prepare readers (instead of making them hang up with their questions). One major issue was: how signal from the head regulates the stem cells, which seems to be everywhere in the body, yet no differential transcripts were found (again, the discussion in the Discussion section was excellent, but none of which were primed in the Results section, so I had to keep reading suspended. Just 'see Discussion' might greatly help the reading).

Reviewer #2:

This paper presents the most comprehensive study of cestode regeneration to date and includes a description of a robust in vitro culture for *Hymenolepis diminuta* that facilitates the use of growth/anatomical bioassays, and powerful techniques like irradiation, cell transplantation and RNAi. Using this in combination with RNAseq, the authors present a fascinating picture of the regenerative capacity of *H. diminuta*, showing that cycling cells from multiple regions of the worm can rescue regeneration in irradiated animals. The RNAseq data also adds valuable resources to the broader flatworm stem cell research community.

This review raises only minor points for clarification and suggests some experimental questions that may warrant consideration/discussion.

Specific comments:

Results, first paragraph: The author should clarify their definition of 'regeneration', especially in the context of planarian 'regeneration'. For example, a head neck and body segment would still constitute a worm with fewer proglottids – so would 'regeneration' in the normal definition be the right word here?

Results, first paragraph: Clarification on the difference between growth and regeneration, and what is actually happening to cause the increase in length, if not regeneration.

Results, first paragraph: Could authors clarify what region of the neck these '2mm "neck only" fragments' came from?

Would it be more correct to refer to *mcm2* and *h2b* as 'proliferative cell markers', rather than 'stem cell markers'?

Results, fourth paragraph: EdU labelling would be visible when positive, even if only a few cells were labelled – could the authors propose alternative hypotheses for new observation of presence of cycling cells in head?

Results, eighth paragraph: Could authors refer to Figure 4B when highlighting the thin and frail worms resulting from the RNAi experiments.

Results, ninth paragraph: Loss of *mcm2* transcript might mean that there are no cycling cells present, but is it possible that the stem cells are still there in a quiescent state?

Results, eleventh paragraph: Should 'gene' be replaced by 'transcript' when discussing RNAseq and ISH?

Clarification of what "subset of cells within neck parenchyma" means. Were the other transcripts not found in the neck or did these 15 genes just show restricted expression in the neck?

Could authors clarify what "but 7/8 genes tested" means?

Results, eleventh paragraph: Does *prox1* not warrant further investigation, or at least discussion?

Results, twelfth paragraph: Although present in the Materials and methods, it would be helpful to reader if the lethal dose was stated here.

Results, twelfth paragraph: Any rationale for 5 mm fragments in this instance considering 2 mm fragments were capable of "regeneration"?

Results, twelfth paragraph: What was the time period between irradiation and injection of cells?

Results, fourteenth paragraph: Although HU concentration is provided in the Materials and methods, again it would be helpful for the reader to state this here.

Clarification of 'posterior donor tissue' – does this means that donor tissues were proglottids?

Discussion, first paragraph: Reference for planarian regeneration?

Subsection “F-ara-EdU31 388 uptake and staining”: For how long was tyramide signal amplification performed? Any difference from planarians?

Subsection “Transcriptome assembly”, third paragraph: RPKM units standardise for length of transcript, so filtering length of transcripts should be unnecessary?

Subsection “RNA-seq for differential gene expression analyses”: Some more detail on exactly how DE analysis was performed would be helpful for reader. Authors refer to expression using RPKM units, although it is common for paired end sequencing data to be referred to using FPKM units.

Other comments:

Did the authors consider the irradiation rescue experiment in decapitated worms?

Did the authors try the irradiation rescue experiment using donor worms having undergone RNAi for one of the cell cycle transcripts (e.g. *h2b*)?

What happens if irradiated worms have cells transplanted into the head or the proglottids, rather than the neck?

Reviewer #3:

The capacity for immense growth and regeneration is a fundamental problem of parasitology. The authors developed the parasitic tapeworm *H. diminuta* as a modern molecular genetic system in an impressive technical advance promoted by an in vitro culture system. The authors found that the neck region of the tapeworm was necessary and sufficient for regeneration of proglottids, but that a head could not be regenerated and was necessary for serial rounds of regeneration. Dividing cells, however, were present in all regions of the body. The authors developed a transplantation procedure that showed cycling cells from multiple regions of the body were capable of rescuing lethally irradiated hosts when transplanted into the neck, indicating that the neck harbors a permissive environment for stem cell proliferation. The novelty in the work involves the comprehensive testing of the regenerative ability of tapeworms, the molecular description of the *H. diminuta* cycling cell population, and in the discovery of the existence of essential proliferating cell-extrinsic anterior cues required for stem cell-driven regeneration.

Results, first paragraph: Please clarify the description of growth without proglottid formation. Show data on "differentiate mature reproductive structures"; there is also a "data not shown" statement about head regeneration which would be better to show.

Some genes were irradiation sensitive and near but not co-expressed with proliferation markers (Figure 3—figure supplement 3D). EdU pulse followed by fixation at different timepoints could support their hypothesis for case study genes that they are expressed in early progeny of cycling cells.

The prominence of signal from gonads makes visualization of proliferating mesenchymal cells difficult in data presented from the posterior. Higher magnification FISH of data such as in Figure 4G or Figure 3A would be helpful.

How far posterior could cells be isolated and still be transplanted and result in successful rescue? The explicit details of the region donor posterior cells came from could be better described, or even further posterior regions could be used in transplants. (i.e., did the cells have to come from near the neck, or is it clear that cells distal to the neck can engraft and support proliferation)?

The authors could more explicitly compare the data obtained about the genes expressed in the cycling cell population of *H. diminuta* to data from neoblasts in planarians (such as *zmym3* and su(Hw) – but ideally systematically with all validated cycling cell markers). A fuller discussion comparing the molecular biology of these cells could add additional depth to the work.

EdU experiments with amputated body fragments could show if posterior cycling cells are capable of producing multiple differentiated cells (with marker double-labeling) in tissue maintenance/growth. This could help in address comments on pluripotency/regeneration models in the Discussion.

---

## [Author Response]

Reviewer #1:[…] Specific comments:- The fourth paragraph of the Results states that the only proliferative somatic cells are undifferentiated in tapeworm, citing Bolla and Roberts, 1971 and Sulgostowska, 1972. But these references are from the 70s, and I am not sure (or more in general, readers will wonder) what kind of methods in the 70s could provide such a conclusion at the accuracy of today's standard. Also, if old studies are that conclusive, this study would sound like not providing much of new insights. Maybe a bit more of elaboration (what was done before, and how it compares to the present study, in providing more accurate information) would be helpful.

We have added a more thorough description as suggested.

“In flatworms, it has been repeatedly shown that the only proliferative cells are undifferentiated cells with stem cell morphology and/or function; these cells have been termed neoblasts, adult somatic stem cells, or germinative cells, depending on the organism (Collins et al., 2013; Koziol et al., 2014; Baguñà, Salo and Auladell, 1989; Ladurner, Rieger and Baguñà, 2000). […] Thus, cycling somatic cells in *H. diminuta* would not include differentiated cells, but would include stem cells and any dividing progeny.”

- Figure 1B: the lack of regeneration with 'body only' is not obvious due to different scales used between day 0 vs. day 9.

We have added Figure 1—figure supplement 1 and text (see below) to describe how the body only fragment increases in length without adding new proglottids. At day 0, the proglottids in the amputated “body only” fragments are small and immature but with time, they grow in size and become reproductively mature. Additionally, since there is no regeneration, they do not add new (and small) proglottids. We show that the mean proglottid length is significantly increased in the “body only” fragments compared to the regeneration-competent fragments. We also show higher magnification images of the most mature proglottids that are observed in the “body only” fragments.

“Despite the failure to regenerate, “body only” fragments could grow because each existing proglottid increased in length as it progressively matured (Figure 1—figure supplement 1A-B).”

- Figure 2 used irradiation to deplete proliferating cells, which the authors assumed to eliminate stem cells. Whereas I agree that this method has been successfully used to identify neoblast-specific genes in planaria, the formal possibility remains that irradiation induce transient quiescence of stem cells. If so, this experiment will be only identifying 'proliferation-associated' genes, instead of 'stem cell specific genes'. In case of planaria, the use of 'lethal dose' (i.e. no stem cells are left indeed) excluded such a possibility. I am guessing 200 Gy for tapeworm is the same treatment, but it is not explained in the paragraph where this experiment is explained (Results, fifth paragraph). Clarification might help here.

200 Gy is a lethal dose of irradiation. 100% of 5 mm anterior fragments amputated from worms exposed to 200 Gy x-irradiation will degenerate after 1 month with no detectable proglottids. We have added a more thorough description to the text and have added supporting data to Figure 2—figure supplement 1.

“Exposing *H. diminuta* to 200 Gy x-irradiation reduced the number of cycling cells by 91 ± 6% after 3 days (Figure 2C-D) and abrogated both growth and regeneration (Figure 2—figure supplement 1A-B). This dosage is lethal; all fragments from worms exposed to 200 Gy x-irradiation degenerated after 1 month (Figure 2—figure supplement 1C-D).”

- Figure 3 explains proliferating cells are in neck parenchyma. Abstract/Introduction primed me/readers to think that stem cells are not limited to neck region and are present everywhere. However, in this figure, the authors' description focuses on neck area (starting in the sixth paragraph of the Results, they provide detailed location of cycling cells within the neck region, without mentioning the other area of the worm-like body), which inevitably made me wonder whether other regions also exhibit similar expression patterns or not, and I was quite confused going through the explanation around here. I wonder these questions might not occur to authors, because the images may be self-explanatory to them. But those who are not familiar to tapeworm anatomy cannot quite tell where the neck is/where the body starts in the presented images (labelling those regions in the panels will greatly help).

We have added annotations to Figure 3B to help highlight that only a portion of the neck is shown. All genes that were expressed in the neck parenchyma were expressed throughout the whole body. We have added images of in situs at the posterior termini in Figure 3—figure supplement 1.

“All transcripts that were expressed in the neck parenchyma were also found throughout the worm body, even in the most posterior proglottids (Figure 3—figure supplement 1B-C).”

- Results, ninth paragraph: Reduction in mcm2 is used to conclude that cell populations are gone after RNAi of h2b etc. However, as mcm2 itself is a cell proliferation gene, it is still possible that the cell population still exists while reducing the expression of mcm2. Here I am not asking to do additional experiments to distinguish these possibilities (cell population is gone vs. proliferation is diminished). I am simply pointing out a flaw in interpretation so that they can adjust their statement.

We have adjusted the description to more accurately describe our observations.

“Are these RNAi-induced failures in growth and regeneration due to defects in the cycling-cell population? RNAi knockdown of *h2b, zmym3,* and *pogzl* severely reduced the number of proliferative cells in the neck that could incorporate F-*ara*-EdU(Figure 4D-E). […] Therefore, *h2b, zmym3,* and *pogzl* are necessary for the maintenance and/or proper function of cycling cells, likely including stem cells, in *H. diminuta*.”

- Figure 5-6 conclude that stem cells are not limited to the neck region. This is based on the lack of any transcripts that are differentially expressed between posterior vs. anterior regions, and the fact that cells from any regions can rescue lethally irradiated animals. Based on these data, the authors propose that head/neck serves to provide extrinsic signals to maintain stem cells, yet there are no intrinsic differences among stem cells. They also nicely show that cycling cells contain the stem cell population (by HU-induced depletion of cycling cells). Whereas the data are striking and clear, the explanation seems to be somewhat confusing (or indicating something is missing). ---upon reading the Discussion, I see that most of the issues (below) are indeed discussed well, but as I read through the Results section, the description went on without addressing some major question. It might be helpful to slip in a few sentences also in Results section to prepare readers (instead of making them hang up with their questions). One major issue was: how signal from the head regulates the stem cells, which seems to be everywhere in the body, yet no differential transcripts were found (again, the discussion in the Discussion section was excellent, but none of which were primed in the Results section, so I had to keep reading suspended. Just 'see Discussion' might greatly help the reading).

We have added a more thorough description in the Results section:

“With this functional assay in hand, we examined the rescue ability of cells from anterior donor tissues (including the regeneration-competent neck) compared to donor tissues from the most posterior termini of 6 day-old tapeworms (which are regeneration incompetent and exclusively comprised of proglottids). […] It appears that in tapeworms, location matters enormously: the head and neck environment provide cues that regulate the ability of stem cells to regenerate proglottids, even though cycling cells (and likely stem cells), are not anatomically confined.”

Reviewer #2:[…] Specific comments:Results, first paragraph: The author should clarify their definition of 'regeneration', especially in the context of planarian 'regeneration'. For example, a head neck and body segment would still constitute a worm with fewer proglottids – so would 'regeneration' in the normal definition be the right word here?

Regeneration is broadly defined as “the replacement of a body part lost through traumatic injury (either amputation or autotomy)” (Bely and Nyberg, 2010). This applies to a broad range of biological levels including regeneration of the whole body, individual anatomical structures, internal organs, tissues and cells. In the first paragraph of the Results section, we describe that we have observed only proglottid regeneration in *H. diminuta* so that the specific regenerative ability demonstrated by these worms is clear.

We have never come across a definition of regeneration that refers to the animal size. If regenerated tapeworms are cultured for long enough they will achieve the same size and number of proglottids as unamputated worms in vitro. Even in the extreme case of whole-body regeneration exhibited by planarians, the regenerated worm will start off smaller than the original worm. Thus, having fewer proglottids than the original tapeworm at any given time after amputation does not negate that the tapeworm can regenerate proglottids following amputation.

Reference:

Bely, A. E., and Nyberg, K. G. (2010). Evolution of animal regeneration: re-emergence of a field. Trends in Ecology and Evolution, 25(3), 161–170.

Results, first paragraph: Clarification on the difference between growth and regeneration, and what is actually happening to cause the increase in length, if not regeneration.

We have added Figure 1—figure supplement 1 and text (see below) to describe how the body only fragment increases in length without adding new proglottids. At day 0, the proglottids in the amputated “body only” fragments are small and immature but with time, they grow in size and become reproductively mature. Additionally, since there is no regeneration, they do not add new (and small) proglottids. We show that the mean proglottid length is significantly increased in the “body only” fragments compared to the regeneration-competent fragments. We also show higher magnification images of the most mature proglottids that are observed in the “body only” fragments.

“Despite the failure to regenerate, “body only” fragments could grow because each existing proglottid increased in length as it progressively matured (Figure 1—figure supplement 1A-B).”

Results, first paragraph: Could authors clarify what region of the neck these '2mm "neck only" fragments' came from?

The neck of 6 day-old tapeworms used in this study is ~2-3 mm in length. We define the neck as the region between the base of the scolex and the first recognizable proglottid, using DAPI staining and fluorescent microscopy. We amputate the head, then cut the following 2 mm neck tissue under a stereomicroscrope. Thus, the “neck only” tissue comprises all/nearly all of the neck tissue. We’ve added text to clarify this.

“The neck of 6 day-old tapeworms used in this study is typically 2-3 mm long when observed after DAPI staining and widefield fluorescent microscopy. By amputating 2 mm “neck only” fragments, we find that the neck is sufficient to regenerate an average of 383 proglottids (SD=138, N=4, n=20) after 12 days in vitro(Figure 1E).”

Would it be more correct to refer to mcm2 and h2b as 'proliferative cell markers', rather than 'stem cell markers'?

In the references cited they are one and the same. But we have no objection to referring to *mcm2* and *h2b* as cycling cells markers and have made the change (Results, fourth paragraph).

Results, fourth paragraph: EdU labelling would be visible when positive, even if only a few cells were labelled – could the authors propose alternative hypotheses for new observation of presence of cycling cells in head?

We believe that Reviewer #2 is asking why previous studies failed to see cycling cells in the head. In 1972, Bolla and Roberts exposed *H. diminuta* to tritiated thymidine for 10 min, stained sections, and exposed the autoradiographs for 2 weeks. The sensitivity of this method allowed them to see the abundant cycling cells in the neck but the scarce cycling cells in the head were probably lost to background signal or missed during sectioning. The negative data is not shown in their paper. It seems likely that due to the technical limitations of these early detection methods, they concluded that there are no cycling cells in the head.

Results, eighth paragraph: Could authors refer to Figure 4B when highlighting the thin and frail worms resulting from the RNAi experiments.

Added (Results, eighth paragraph).

Results, ninth paragraph: Loss of mcm2 transcript might mean that there are no cycling cells present, but is it possible that the stem cells are still there in a quiescent state?

We have adjusted the description to more accurately describe our observations.

“Are these RNAi-induced failures in growth and regeneration due to defects in the cycling-cell population? RNAi knockdown of *h2b, zmym3,* and *pogzl* severely reduced the number of proliferative cells in the neck that could incorporate F-*ara*-EdU(Figure 4D-E). […] Therefore, *h2b, zmym3,* and *pogzl* are necessary for the maintenance and/or proper function of cycling cells, likely including stem cells, in *H. diminuta*.”

Results, eleventh paragraph: Should 'gene' be replaced by 'transcript' when discussing RNAseq and ISH?

We have made these substitutions throughout the manuscript.

Clarification of what "subset of cells within neck parenchyma" means. Were the other transcripts not found in the neck or did these 15 genes just show restricted expression in the neck?Could authors clarify what "but 7/8 genes tested" means?

These 15 transcripts showed expression within the neck parenchyma but only in a subset of cells (Figure 5C). Other anterior-enriched transcripts were broadly expressed in the parenchyma/weak/not predominantly in the neck parenchyma. Of the 15 transcripts, we obtained unambiguous and clear results for 8 dFISH experiments. We have clarified our description in the main text:

“We found 15 transcripts expressed in a subset of cells within the neck parenchyma (Figure 5C) and initially hypothesized that these transcripts may represent subsets of stem cells. We were able to successfully test 8 candidates by dFISH with cycling-cell markers and found that the majority (7/8) were not expressed in cycling cells (Figure 5D, Supplementary file 1B).”

Results, eleventh paragraph: Does prox1 not warrant further investigation, or at least discussion?

We do not have enough data to conclude or speculate about the *prox1+* cells though we hope that future experiments will help elucidate their identity and function. RNAi of *prox1* did not reveal strong gross morphological defects. If the *prox1+* cells are lineage committed stem cells/progenitors, then it is likely no gross defects occur but that loss of specific lineages could be determined if we knew more about the identity of these cells. Furthermore, we are reluctant to interpret this negative result especially since RNAi will only reduce and not eliminate *prox1* expression.

We added text to acknowledge that the identity and function of these cells is currently unknown, as well as to emphasize that even *prox1* is not expressed in a neck-restricted fashion.

“At present, the identity and function of *prox1^+^* cells is unknown. Furthermore, *prox1* is expressed throughout the tapeworm body (Figure 3—figure supplement 1).”

Results, twelfth paragraph: Although present in the Materials and methods, it would be helpful to reader if the lethal dose was stated here.

We have added the dosage (Results, thirteenth paragraph).

Results, twelfth paragraph: Any rationale for 5 mm fragments in this instance considering 2 mm fragments were capable of "regeneration"?

We chose to amputate larger fragments to allow for rescue to occur before too much tissue degeneration happened. Having said that, we did not explicitly test if the rescue would be successful in fragments smaller than 5 mm. Since all 5 mm fragments that were irradiated with a lethal dose degenerated and had no proglottids after 30 days, we deemed this protocol suitable.

Results, twelfth paragraph: What was the time period between irradiation and injection of cells?

Worms were irradiated and then injected with cells on the same day, as soon as the dissociated cell preps were ready. We have added this clarification to the Materials and methods section:

“For all rescue experiments, cells were injected into irradiated hosts on the same day that the hosts were irradiated.”

Results, fourteenth paragraph: Although HU concentration is provided in the Materials and methods, again it would be helpful for the reader to state this here.

Added (Results, fifteenth paragraph).

Clarification of 'posterior donor tissue' – does this means that donor tissues were proglottids?

Yes. We took 5 mm of the most posterior termini of 6 day-old tapeworms, which are exclusively comprised of proglottids. We have added this description in the text.

“With this functional assay in hand, we examined the rescue ability of cells from anterior donor tissues (including the regeneration-competent neck) compared to donor tissues from the most posterior termini of 6 day-old tapeworms (which are regeneration incompetent and exclusively comprised of proglottids).”

Discussion, first paragraph: Reference for planarian regeneration?

Added (Discussion, first paragraph).

Subsection “F-ara-EdU31 388 uptake and staining”: For how long was tyramide signal amplification performed? Any difference from planarians?

10-20 min depending on the size of the tissue. This is similar to planarians. We have added the development time to the Materials and methods section.

Subsection “Transcriptome assembly”, third paragraph: RPKM units standardise for length of transcript, so filtering length of transcripts should be unnecessary?

This was not part of the differential gene expression analysis. This is part of the transcriptome assembly. We used a length filter to reduce potential spuriously assembled contigs if they did not also meet the criteria described in the Materials and methods.

Subsection “RNA-seq for differential gene expression analyses”: Some more detail on exactly how DE analysis was performed would be helpful for reader. Authors refer to expression using RPKM units, although it is common for paired end sequencing data to be referred to using FPKM units.

We have added more description of the DGE analysis, which we performed using the recommended standards in CLC Genomics Workbench 6. Estimated tagwise dispersions were calculated using total read counts after mapping to the transcriptome.

“Paired-end reads were mapped to the transcriptome (above) using default settings on CLC Genomics Workbench 6 (Qiagen) except that read alignments were done with a relaxed length fraction of 0.5. Differential gene expression analysis was done with the same software using estimate tagwise dispersions on total read counts and a total count filter cut-off of 5 reads. All sequence reads used for differential genes expression analyses are available at GenBank Bioproject PRJNA546293.”

Other comments:Did the authors consider the irradiation rescue experiment in decapitated worms?

After decapitation, the neck is not maintained and the entire tissue becomes proglottidized. Since donor cells from proglottids alone could already rescue lethally irradiated worms, we did not pursue this specific experiment.

Did the authors try the irradiation rescue experiment using donor worms having undergone RNAi for one of the cell cycle transcripts (e.g. h2b)?

The cell dissociation protocol we used in this study was very harsh in order to overcome the integrity of the tegument. As a consequence, very few cycling cells were incorporated despite performing bulk cell transplants (Figure 6—figure supplement 1). Since RNAi of cycling cell transcripts like *h2b* results in extremely small worms, it would take an enormous number of RNAi worms to perform a rescue experiment. We hope to overcome these technical hurdles with future optimization, but currently, this experiment is too technically challenging.

What happens if irradiated worms have cells transplanted into the head or the proglottids, rather than the neck?

We are currently pursuing these kinds of experiments but they are beyond the scope of this paper.

Reviewer #3:[…] Results, first paragraph: Please clarify the description of growth without proglottid formation. Show data on "differentiate mature reproductive structures"; there is also a "data not shown" statement about head regeneration which would be better to show.

We have added Figure 1—figure supplement 1 and text (see below) to describe how the body only fragment increases in length without adding new proglottids. At day 0, the proglottids in the amputated “body only” fragments are small and immature but with time, they grow in size and become reproductively mature. Additionally, since there is no regeneration, they do not add new (and small) proglottids. We show that the mean proglottid length is significantly increased in the “body only” fragments compared to the regeneration-competent fragments. We also show higher magnification images of the most mature proglottids that are observed in the “body only” fragments.

“Despite the failure to regenerate, “body only” fragments could grow because each existing proglottid increased in length as it progressively matured (Figure 1—figure supplement 1A-B).”

We have also added data of head fragment regeneration failure (Figure 1—figure supplement 1).

“Furthermore, amputated heads alone could not regenerate in vitro (Figure 1—figure supplement 1C) nor in vivo (Read, 1967).”

Some genes were irradiation sensitive and near but not co-expressed with proliferation markers (Figure 3—figure supplement 3D). EdU pulse followed by fixation at different timepoints could support their hypothesis for case study genes that they are expressed in early progeny of cycling cells.

Unfortunately, we are currently unable to perform in situs in conjunction with F-*ara*-EdU staining. F-*ara*-EdU is toxic to tapeworms at concentrations above 1 μM (we use 0.1 μM). In order to detect F-*ara*-EdU, we need extensive tissue permeabilization (several days in PBSTx at room temperature, proteinase K digestion, and DMSO+detergents) before the click-it reaction. Even then, the signal is only clearly visible after antibody amplification and TSA reaction. Performing the F-*ara*-EdU staining protocol after our in situ protocol compromises both the in situ and the F-*ara*-EdU signals. At this stage, this is a technical limitation we have not overcome.

The prominence of signal from gonads makes visualization of proliferating mesenchymal cells difficult in data presented from the posterior. Higher magnification FISH of data such as in Figure 4G or Figure 3A would be helpful.

We have added higher magnification confocal sections of in situs from the animal posterior in Figure 4—figure supplement 2. We have circled the gonads so that the mesenchymal expression of these genes is more obvious.

“By WISH and FISH, all cycling-cell transcripts including *zmym3* and *pogzl* were detected throughout the whole tapeworm body(Figure 4G, Figure 3—figure supplement 1B-C). […] Since *zmym3* and *pogzl* label all cycling cells, it is possible that stem cells of limited potential exist in the posterior, but an elusive subpopulation of pluripotent stem cells is confined to the neck.”

How far posterior could cells be isolated and still be transplanted and result in successful rescue? The explicit details of the region donor posterior cells came from could be better described, or even further posterior regions could be used in transplants. (i.e., did the cells have to come from near the neck, or is it clear that cells distal to the neck can engraft and support proliferation)?

We have added more description of the posterior donor tissue. The tissue used was the most posterior termini comprised exclusively of proglottids. Thus, the cells used were the most distal from the neck at the time (6 day-old worms).

“With this functional assay in hand, we examined the rescue ability of cells from anterior donor tissues (including the regeneration-competent neck) compared to donor tissues from the most posterior termini of 6 day-old tapeworms (which are regeneration incompetent and exclusively comprised of proglottids).”

The authors could more explicitly compare the data obtained about the genes expressed in the cycling cell population of H. diminuta to data from neoblasts in planarians (such as zmym3 and su(Hw) – but ideally systematically with all validated cycling cell markers). A fuller discussion comparing the molecular biology of these cells could add additional depth to the work.

As per the reviewer’s suggestion, we have added a new Supplementary file 1C in which we analyze the verified tapeworm cycling cell markers against planarian neoblast genes described in three studies (Fincher et al., 2018; Plass et al., 2018; Labbe et al., 2012). This analysis was also facilitated by the planarian resource Planmine. We have added text to the Results and Discussion.

Results:

“The transcriptional heterogeneity detected in the cycling-cell compartment is reminiscent of similar observations made in the regenerative planarian *S. mediterranea*. A comparative analysis between verified tapeworm cycling-cell transcripts and their putative planarian homologs revealed a number of transcripts with conserved expression in cycling-cell populations from these distantly related flatworms (Supplementary file 1C) (see Discussion).”

Discussion:

“How do the cycling-cell transcripts we identified in *H. diminuta* compare to stem cells in free-living planarians? (Fincher et al., 2018; Plass et al., 2018; Labbé et al., 2012; Rozanski et al., 2019) (Supplementary file 1C). […] Thus, despite >500 million years of separation between free-living and parasitic flatworm evolution (Laumer, Hejnol and Giribet, 2015), tapeworm cycling-cell transcripts have conserved signatures with planarian neoblasts.”

In refence to the potential molecular functions of *zmym3* and *pogzl*, we have added text to the Discussion.

“Both *zmym3* and *pogzl* are neoblast cluster-defining genes in planarians (Supplementary file 1C) suggesting that their functions in stem cell regulation may be conserved across the two species. […] Thus, it would be interesting to further understand the mechanism of action of *zmym3* and *pogzl* in stem cells of parasitic and free-living flatworms.”

EdU experiments with amputated body fragments could show if posterior cycling cells are capable of producing multiple differentiated cells (with marker double-labeling) in tissue maintenance/growth. This could help in address comments on pluripotency/regeneration models in the Discussion.

As described above, F-*ara*-EdU in combination with in situ hybridizations is currently not feasible. However, we were able to do pulse-chase experiments and use antibody staining as well as anatomical location to identify differentiated cells: anti-acetylated α- tubulin antibodies label flame cells of the protonephridial system and the edge-most nuclei are differentiated muscle and tegument. We were able to pulse amputated posterior fragments of proglottids alone (2mm) and perform 1 hr F-*ara*-EdU pulse followed by a 3 days chase and found that posterior cycling cells could chase into muscle/tegument and flame cells. The data are presented in Figure 6—figure supplement 2 and in the text.

“Interestingly, using pulse-chase experiments with F-*ara*-EdU, we find that the cycling cells of posterior proglottids can give rise to multiple differentiated cell types like muscle/tegument at the animal edge as well as flame cells of the protonephridial system marked by anti-acetylated α-tubulin antibodies (Rozario and Newmark, 2015) (Figure 6—figure supplement 2). Thus, the cycling cells from tapeworm posteriors show hallmarks of stem cell activity, despite the fact that this tissue is not competent to regenerate.”